# IMPROVED FINITE-PARTICLE CONVERGENCE RATES FOR STEIN VARIATIONAL GRADIENT DESCENT

**Sayan Banerjee**
Department of Statistics and Operations Research
University of North Carolina, Chapel Hill
sayan@email.unc.edu

**Krishnakumar Balasubramanian**
Department of Statistics
University of California, Davis
kbala@ucdavis.edu

**Promit Ghosal**
Department of Statistics
University of Chicago
promit@uchicago.edu

## ABSTRACT

We provide finite-particle convergence rates for the Stein Variational Gradient Descent (SVGD) algorithm in the Kernelized Stein Discrepancy (KSD) and Wasserstein-2 metrics. Our key insight is that the time derivative of the relative entropy between the joint density of $N$ particle locations and the $N$-fold product target measure, starting from a regular initial distribution, splits into a dominant 'negative part' proportional to $N$ times the expected $\mathsf{KSD}^2$ and a smaller 'positive part'. This observation leads to $\mathsf{KSD}$ rates of order $1/\sqrt{N}$, in both continuous and discrete time, providing a near optimal (in the sense of matching the corresponding i.i.d. rates) double exponential improvement over the recent result by Shi & Mackey (2024). Under mild assumptions on the kernel and potential, these bounds also grow polynomially in the dimension $d$. By adding a bilinear component to the kernel, the above approach is used to further obtain Wasserstein-2 convergence in continuous time. For the case of 'bilinear + Matérn' kernels, we derive Wasserstein-2 rates that exhibit a curse-of-dimensionality similar to the i.i.d. setting. We also obtain marginal convergence and long-time propagation of chaos results for the time-averaged particle laws.

## 1 INTRODUCTION

Stein Variational Gradient Descent (SVGD) (Liu & Wang, 2016) is a widely-used deterministic particle-based algorithm for sampling from a target density $\pi \propto \exp(-V)$, where $V : \mathbb{R}^d \to \mathbb{R}$ is the potential function. For a given symmetric, positive-definite kernel $\mathsf{k} : \mathbb{R}^d \times \mathbb{R}^d \to \mathbb{R}$, discrete time-step $n \in \mathbb{N}_0$, step-size $\eta > 0$, and for $1 \le i \le N$, the SVGD algorithm is given by

$$x_i^N(n+1) = x_i^N(n) - \frac{\eta}{N} \sum_j \left[ \mathsf{k}(x_i^N(n), x_j^N(n)) \nabla V(x_j^N(n)) - \nabla_2 \mathsf{k}(x_i^N(n), x_j^N(n)) \right]. \quad (1)$$

SVGD provides a compelling alternative to more classical randomized sampling algorithms like Markov Chain Monte Carlo (MCMC) that require additional uncertainty quantification with respect to the algorithmic randomness. It has attracted considerable attention in the machine learning and applied mathematics communities because of its fascinating theoretical properties and broad range of applications (Feng et al., 2017; Haarnoja et al., 2017; Lambert et al., 2021; Liu et al., 2021; Xu et al., 2022). Our focus in this work is on deriving rates of convergence of the SVGD algorithm in equation 1 and the corresponding continuous-time, $N$-particle SVGD dynamics on $\mathbb{R}^d$, obtained by letting $\eta \to 0_+$, given by

$$\dot{x}_i^N(t) = -\frac{1}{N} \sum_j \mathsf{k}(x_i^N(t), x_j^N(t)) \nabla V(x_j^N(t)) + \frac{1}{N} \sum_j \nabla_2 \mathsf{k}(x_i^N(t), x_j^N(t)), \quad (2)$$

with $\dot{x}$ denoting the time derivative and $\nabla_2$ represents gradient with respect to the second argument. Throughout the paper, we will implicitly assume the existence of a solution to the above equation and the completeness of the vector field driving the above dynamics in $\left(\mathbb{R}^d\right)^N$. This is satisfied, for example, under standard continuity and linear growth assumptions on the driving vector field.

The motivation for SVGD originates from the *gradient flow for the relative entropy (i.e., the* KL *divergence)* on the Wasserstein-2 space of probability measures on $\mathbb{R}^d$. More precisely, for a probability measure $\mu$ on $\mathbb{R}^d$ possessing a regular enough positive density, the Wasserstein gradient flow is given by the measure-valued trajectory $\mu_t$ satisfying the continuity equation

$$\partial_t \mu_t + \nabla \cdot (v_t \mu_t) = 0, \quad \mu_0 = \mu, \tag{3}$$

where $v_t = -\nabla \log(p_t/\pi)$ and $p_t$ is the density of $\mu_t$. Under suitable conditions, $\mu_t$ can be shown to converge (often with quantifiable fast rates) to $\pi$. Unfortunately, this approach is not practically implementable via particle discretization as the associated empirical measure approximating $\mu_t$ does not possess a density.

In a very influential paper, Liu & Wang (2016) devised a *projected* gradient descent algorithm by projecting the velocity vector $v_t$ along a reproducing kernel Hilbert space (RKHS) associated with a symmetric positive definite kernel $\mathsf{k}$. This leads to a flow analogous to equation 3 but with $v_t = -P_{\mu_t} \nabla \log(p_t/\pi)$, where the projection $P_{\mu_t}$ is given by $P_\nu f(x) := \int \mathsf{k}(x,y) f(y) \nu(\mathsf{d}y)$ for probability measure $\nu$ and function $f : \mathbb{R}^d \to \mathbb{R}$ for which the integral is well-defined. The key observation of Liu & Wang (2016) was that, by applying integration by parts, one obtains

$$-P_{\mu_t} \nabla \log(p_t/\pi)(x) = \int \left( -k(x,y) \nabla V(y) + \nabla_2 k(x,y) \right) \mu_t(\mathsf{d}y).$$

The right hand side is well-defined even when $\mu_t$ lacks a density and is hence amenable to particle discretization, which leads to the SVGD equations equation 1 and equation 2. See Korba et al. (2020) for a more detailed description of this approach.

**Challenges for finite-particle SVGD:** There has been extensive work in quantifying convergence rates for the mean-field SVGD equation (see '**Past works**' below). However, only Shi & Mackey (2024) and Liu et al. (2024) have made attempts towards obtaining rates for the finite-particle version of (deterministic) SVGD. This has been perceived as a challenging open problem till date. The tractability of the mean-field SVGD equation comes from the observation that it has a (projected) gradient structure which leads to the following monotonicity property of the KL-divergence:

$$\partial_t \mathsf{KL}(\mu_t || \pi) = -\mathsf{KSD}^2(\mu_t || \pi), \quad t \geq 0,$$

where KSD stands for the Kernelized Stein Discrepancy (Chwialkowski et al. (2016); Liu et al. (2016); Gorham & Mackey (2017)). The non-negativity of KL then leads to bounds on the KSD. For the finite-particle versions equation 1 and equation 2, there is no gradient structure to the dynamics, which renders the above approach inapplicable. Moreover, the vector field driving the finite-particle dynamics is not globally Lipschitz and lacks suitable convexity properties. This results in double-exponentially growing bounds in time between the particle empirical distribution and the mean-field limit (see Lu et al. (2019, Prop. 2.6) and Shi & Mackey (2024, Thm. 1)). As a consequence, attempts to demonstrate finite-particle convergence to the target distribution $\pi$ by relying on the mean-field convergence and the convergence of the mean-field equation to the target in the general (non-Gaussian) setting lead to a slow convergence rate of $1/\sqrt{\log \log N}$. In Das & Nagaraj (2023), the authors intentionally bypassed this approach for the finite-particle setting, achieving improved convergence rates, but their method requires a distinct, albeit related, algorithm that incorporates additional randomness into the dynamics.

**Our contributions:** A key insight in this paper is to work with the *joint density* of the particle locations, when started from a suitably regular initial distribution, and track the evolution of its relative entropy with respect to the $N$-fold product measure $\pi^{\otimes N}$. It turns out that the time derivative of this relative entropy has a 'negative part' that is exactly $N$ times the expected $\mathsf{KSD}^2$ of the empirical measure at time $t$ with respect to $\pi$, and a 'positive part' that can be separately handled and shown to be small in comparison to the negative part (see equation 9). This gives a novel connection between the joint particle dynamics and the empirical measure evolution.

Our first main result, Theorem 1, exploits this observation to obtain $O(1/\sqrt{N})$ bounds for the expected KSD between $\mu_{av}^N := \frac{1}{N} \int_0^N \mu^N(t) \mathsf{d}t$ and $\pi$ for the continuous-time SVGD dynamics in

equation 2. Analogous bounds for the discrete-time SVGD dynamics in equation 1 are obtained in Theorem 3. Together, these results constitute a *double exponential improvement over Shi & Mackey (2024) for the true SVGD algorithm*. As discussed in Remark 2, the bounds in Theorem 1 are *essentially optimal* when compared with the KSD in the i.i.d. setting, and *grow linearly in $d$* (that is, KSD is $O(d/\sqrt{N})$) under mild assumptions on the kernel and the potential. Moreover, unlike previous works even for the mean-field SVGD, we do not require any assumptions on the tail behavior (such as sub-Gaussianity) of the target $\pi$ in Theorem 1. Further, it follows from Gorham & Mackey (2017) that the KSD bound alone does not even guarantee weak convergence of the particle marginal laws as $N \to \infty$, unless one establishes tightness of these laws. Our approach gives control on the relative entropy of the joint law in time which, in turn, gives the desired tightness and weak convergence for the time-averaged particle marginal laws $\bar{\mu}^N(\cdot) := \frac{1}{N} \int_0^N \mathbb{P}(x_1(t) \in \cdot) \mathrm{d}t$, for exchangeable initial conditions, see Theorem 2.

The discrete-time SVGD bound in Theorem 3, although similar in flavor to Theorem 1, is substantially more involved and requires careful control on the discretization error. In this result, we incorporate a parameter $\alpha$ that lets us interpolate between exponential tails and Gaussian tails as $\alpha$ varies from 0 to $1/2$. This parameter turns out to be crucial in choosing the step-size $\eta$, which is $\approx d^{-(\frac{1+\alpha}{2(1-\alpha)} \vee 1)} N^{-\frac{1+\alpha}{1-\alpha}}$, and the number of iterations required $\approx N^{\frac{2}{1-\alpha}}$, to obtain KSD bounds which are $O\left(d^{(\frac{3-\alpha}{4(1-\alpha)} \vee 1)} N^{-1/2}\right)$. Unlike the population limit discrete-time SVGD rates previously obtained in Korba et al. (2020); Salim et al. (2022), the Hilbert-Schmidt norm of the Jacobian of the transformation associated with each iteration depends non-trivially on the initial configuration and the number of iterations. A key technical ingredient in controlling this is an 'a priori' bound on the functional $n \mapsto N^{-1} \sum_i V(x_i^N(n))$ obtained in Lemma 3.

In Section 4, we obtain *Wasserstein-2 convergence and associated rates*. For this purpose, we heavily rely on the treatise of Kanagawa et al. (2022) which connects KSD convergence to Wasserstein convergence when the kernel has a bilinear component and a translation invariant component of the form $(x, y) \mapsto \Psi(x - y)$ (see equation 11). Such kernels are typically unbounded, in contrast with standard boundedness assumptions in most papers on SVGD (a notable exception is Liu et al. (2024)). In Theorem 4, under dissipativity and growth assumptions on the potential $V$, we obtain polynomial KSD convergence rates for SVGD finite-particle dynamics with such kernels, which by Kanagawa et al. (2022) imply Wasserstein convergence. When the translation invariant part of the kernel is of Matérn type, we obtain Wasserstein convergence rates in Theorem 5 of the form $O(1/N^{\alpha/d})$ (where $\alpha > 0$ does not depend on $d$) for the particle SVGD using Theorem 4 in conjunction with results in Kanagawa et al. (2022). This is the first work on Wasserstein convergence for non-Gaussian SVGD finite-particle dynamics. Unlike the KSD bound, the $d$ dependence leads to *curse-of-dimensionality* in the Wasserstein bound, but this is to be expected when compared to Wasserstein bounds for empirical distribution of i.i.d. random variables (Dudley, 1969; Weed & Bach, 2019). Finally, we obtain a *long-time propagation of chaos* result in Proposition 1, namely, we show that the time-averaged marginals of the particle locations over the time interval $[0, N]$, started from an exchangeable initial configuration, become asymptotically independent as $N \to \infty$ and essentially produce i.i.d samples from $\pi$. Although the results in Sections 4 and 5 are proved for the continuous-time SVGD in equation 2 to highlight the main ideas, analogous results can also be proved in discrete-time and is deferred to future work.

**Past works:** The following diagram from Liu et al. (2024) highlights the major approaches undertaken in rigorously analyzing the SVGD dynamics:

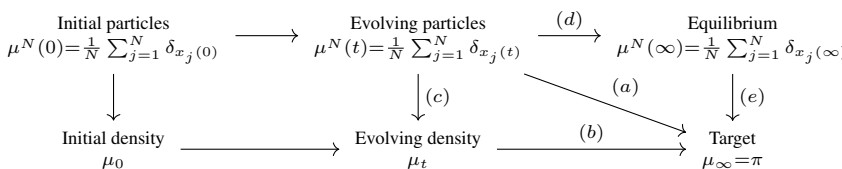

(a) Unified convergence of the empirical measure for $N < \infty$ particles to the continuous target as time $t$ and $N$ jointly grow to infinity;
(b) Convergence of mean-field SVGD to the target distribution over time;

    (c) Convergence of the empirical measure for finite particles to the mean-field distribution at any finite given time $t \in [0, \infty)$;

    (d) Convergence of finite-particle SVGD to equilibrium over time;

    (e) Convergence of the empirical measure for finite particles to the continuous target at time $t = \infty$.

Practically speaking, (a) is the ideal outcome that completely defines the algorithmic behavior of SVGD. One approach towards this is to combine either (b) and (c) or (d) and (e) in a quantitative way to yield (a). Regarding (b), Liu (2017) showed the convergence of mean-field SVGD (solution to equation 3 with $v_t = -P_{\mu_t}\nabla \log(p_t/\pi)$) in KSD which is known to imply weak convergence under appropriate assumptions. Korba et al. (2020); Chewi et al. (2020); Salim et al. (2022); Sun et al. (2023); Duncan et al. (2023) sharpened the results with weaker conditions or explicit rates. He et al. (2024) extended the above result to the stronger Fisher information metric and Kullback–Leibler divergence based on a regularization technique. Lu et al. (2019); Gorham et al. (2020); Korba et al. (2020) obtained time-dependent mean-field convergence (c) under various assumptions using techniques from partial differential equations and from the literature of 'propagation of chaos'. In particular, Lu et al. (2019) derived the mean-field PDE equation 3 for the evolving density that emerges as the mean-field limit of the finite-particle SVGD systems, and showed the well-posedness of the PDE solutions. Carrillo & Skrzeczkowski (2023) established refined stability estimates in comparison to Lu et al. (2019) for the mean-field system when the initial distribution is close to the target distribution in a suitable sense. In particular, they increase the length of the time interval in which mean-field approximation is meaningful from $\approx \log \log N$ to $\approx \sqrt{N}$ for such initial data close to the target.

Shi & Mackey (2024) obtained refined results for (c) and combined them with (b) to get the *first unified convergence* (a) in terms of KSD. However, they have a rather slow rate of order $1/\sqrt{\log \log N}$, resulting from the fact that their bounds for (c) still depend on the time $t$ (sum of step sizes) double-exponentially. Note that studying the convergence (d) and (e), provides another way to characterize the unified convergence (a) for SVGD. Liu et al. (2024) analyzed this strategy for the *Gaussian SVGD* case where the target distribution $\pi$ and initial distribution $\mu$ are both Gaussian and the kernel k is bilinear. In this case, the flow of measures for the mean-field SVGD remains Gaussian for all time and this fact was exploited to obtain detailed rates and 'uniform-in-time' propagation of chaos results. Das & Nagaraj (2023) obtained a polynomial convergence rate ($O(N^{-\alpha})$ for some $\alpha > 0$) for a related but different algorithm, which they called *SVGD with virtual particles*, by adding more randomness to the dynamics and using stochastic approximation techniques. The recent work of Priser et al. (2024) also studies finite-particle asymptotics, albeit for not the original SVGD iterates (as in equation 1) but for a modified one where a Langevin-type regularization including a Gaussian noise is added at each step. Hence, they leverage existing techniques for Langevin Monte Carlo to establish their results. However, their techniques are not applicable to the deterministic SVGD system in equation 1.

**Notation:** We will say a function $f$ is $\mathcal{C}^k$ if it is $k$ times continuously differentiable in its arguments. $\mathcal{L}(X)$ will denote the law of the random variable $X$. We let $\mathcal{B}(\mathbb{R}^d)$ denote the Borel sigma-algebra on $\mathbb{R}^d$. We use $\pi^{\otimes N}$ N-fold product target measure, i.e., $\pi^{\otimes N}(x_1, \ldots, x_N) := \pi(x_1) \times \cdots \times \pi(x_N)$. Throughout the article (particularly in the proofs), we will often suppress the superscript $N$ for various objects when it is clear from context. Furthermore, underlined vectors (e.g., $\underline{x}$) denote objects in $\left(\mathbb{R}^d\right)^N$.

## 2    CONTINUOUS-TIME FINITE-PARTICLE CONVERGENCE RATES IN KSD METRIC

We first provide rates in the KSD metric. Let $\mathcal{H}_0$ denote the reproducing kernel Hilbert space (RKHS) of real-valued functions associated with the positive definite kernel k (Aronszajn, 1950). Then $\mathcal{H} := \mathcal{H}_0 \times \cdots \times \mathcal{H}_0$ inherits a natural RKHS structure comprising $\mathbb{R}^d$-valued functions. The *Langevin-Stein operator* (Gorham & Mackey, 2015) $\mathcal{T}_\pi$, associated with $\pi \propto e^{-V}$, acts on differentiable functions $\phi : \mathbb{R}^d \to \mathbb{R}^d$ by

$$\mathcal{T}_\pi \phi(x) := -\nabla V(x) \cdot \phi(x) + \nabla \cdot \phi(x), \quad x \in \mathbb{R}^d.$$

The *Kernelized Stein Discrepancy*[1] (KSD) (Chwialkowski et al., 2016; Gorham & Mackey, 2017), associated with the kernel $\mathsf{k}$, of a probability measure $P$ on $\mathbb{R}^d$ with respect to $\pi$ is defined as

$$\mathsf{KSD}(P||\pi) \coloneqq \sup\{\mathbb{E}\left[\mathcal{T}_\pi\phi(X)\right]: \ X \sim P, \ \phi \in \mathcal{H}, \|\phi\|_{\mathcal{H}} \leq 1\}. \tag{4}$$

The definition of KSD is motivated by *Stein's identity* which says that, for any sufficiently regular $\phi$, $\mathbb{E}_{X\sim\pi}\left[\mathcal{T}_\pi\phi(X)\right] = 0$ and thus, the above measures the 'distance' of $P$ from $\pi$ via the maximum discrepancy of this expectation from 0 when $X \sim P$, as $\phi$ varies over $\mathcal{H}$.

The appeal of KSD lies in the fact that, unlike most distances on the space of probability measures, KSD has an explicit tractable expression. The function $\phi^* \in \mathcal{H}$ for which the above supremum is attained has a closed form expression $\phi^*(x) \propto \mathbb{E}_{Y\sim P}\left[-\mathsf{k}(Y,x)\nabla V(Y) + \nabla_1\mathsf{k}(Y,x)\right]$. Using this, we get the following expression for KSD:

$$\mathsf{KSD}^2(P||\pi) = \mathbb{E}_{(X,Y)\sim P\otimes P}\left[\nabla V(X) \cdot (\mathsf{k}(X,Y)\nabla V(Y)) - \nabla V(X) \cdot \nabla_2\mathsf{k}(X,Y)\right.$$
$$\left. -\nabla V(Y) \cdot \nabla_1\mathsf{k}(X,Y) + \nabla_1 \cdot \nabla_2\mathsf{k}(X,Y)\right]. \tag{5}$$

Before proceeding, we introduce the following regularity conditions, and state an existence and regularity result (proved in Appendix A.1) for the joint particle density.

**Assumption 1.** *We make the following regularity assumptions.*

  *(a) The maps $(x,y) \mapsto \mathsf{k}(x,y)$ and $x \mapsto V(x)$ are $\mathcal{C}^3$.*

  *(b) $\underline{x}^N(0) = (x_1^N(0), \ldots, x_N^N(0))$ has a $\mathcal{C}^2$ density $p_0^N$.*

**Lemma 1.** *Consider the SVGD dynamics equation 2 under Assumption 1. Then the particle locations $(x_1(t), \ldots, x_N(t))$ have a joint density $p^N(t, \cdot)$ for every $t \geq 0$, and the map $(t, \underline{z}) \mapsto p^N(t, \underline{z})$ is $\mathcal{C}^2$.*

The proof of this lemma is deferred to Appendix A.1. Now we proceed to bound KL-divergence between the joint density of $N$-many particles at time $t$ and the $N$-fold product measure of the target distribution $\pi$.

Denote the KL-divergence as

$$\mathsf{KL}(p^N(t)||\pi^{\otimes N}) \coloneqq \int \log\left(\frac{p^N(t, \underline{z})}{\pi^{\otimes N}(\underline{z})}\right) p^N(t, \underline{z})\mathrm{d}\underline{z}. \tag{6}$$

The following theorem furnishes the key bound on the KSD between the empirical law

$$\mu^N(t) \coloneqq \frac{1}{N}\sum_{i=1}^N \delta_{x_i(t)}, \qquad t \geq 0,$$

and the target distribution $\pi$. Define

$$C^*(z) \coloneqq \nabla_2\mathsf{k}(z,z) \cdot \nabla V(z) + \mathsf{k}(z,z)\Delta V(z) - \Delta_2\mathsf{k}(z,z), \quad z \in \mathbb{R}^d. \tag{7}$$

In the above, $\nabla_2\mathsf{k}(z,z) \coloneqq \nabla_2\mathsf{k}(z,\cdot)(z)$ and $\Delta_2\mathsf{k}(z,z) \coloneqq \Delta_2\mathsf{k}(z,\cdot)(z)$.

**Theorem 1.** *Let Assumption 1 hold. Then, we have for every $T > 0$,*

$$\frac{1}{T}\int_0^T \mathbb{E}[\mathsf{KSD}^2(\mu^N(t)||\pi)]\mathrm{d}t \leq \frac{\mathsf{KL}(p^N(0)||\pi^{\otimes N})}{NT} + \frac{1}{N^2T}\int_0^T \mathbb{E}\left[\sum_{k=1}^N C^*\left(x_k(t)\right)\right]\mathrm{d}t,$$

*where the expectation is with respect to $p(t)$. In addition, we have that*

$$\mathsf{KL}(p^N(T)||\pi^{\otimes N}) \leq \mathsf{KL}(p^N(0)||\pi^{\otimes N}) + \frac{1}{N}\int_0^T \mathbb{E}\left[\sum_{k=1}^N C^*\left(x_k(t)\right)\right]\mathrm{d}t.$$

*Moreover, if $C^* \coloneqq \sup_{z\in\mathbb{R}^d} C^*(z) < \infty$ and $\limsup_{N\to\infty} \mathsf{KL}(p^N(0)||\pi^{\otimes N})/N < \infty$, then*

$$(\mathbb{E}[\mathsf{KSD}(\mu_{av}^N||\pi)])^2 \leq \frac{1}{N}\int_0^N \mathbb{E}[\mathsf{KSD}^2(\mu^N(t)||\pi)]\mathrm{d}t \leq \frac{\sup_L \frac{\mathsf{KL}(p^L(0)||\pi^{\otimes L})}{L} + C^*}{N}, \tag{8}$$

*where $\mu_{av}^N(\mathrm{d}x) \coloneqq \frac{1}{N}\int_0^N \mu^N(t, \mathrm{d}x)\mathrm{d}t.$*

---
[1]See Barp et al. (2022, Section 2.2) for additional technical details regarding the well-definedness of KSD.

**Remark 1.** *The condition $C^* < \infty$ holds, for example, when $\mathsf{k}(u, v) = \Psi(u - v)$ for a positive-definite $\mathcal{C}^3$ function $\Psi : \mathbb{R}^d \to \mathbb{R}$ (Bochner, 1933), and $\sup_{x \in \mathbb{R}^d} \Delta V(x) < \infty$. Examples of such kernels include the radial basis kernel (e.g., Gaussian) and a wide class of Matérn kernels. The condition on the potential allows for a large class of non-log-concave densities as well. The condition $\limsup_{N \to \infty} \mathsf{KL}(p^N(0) \| \pi^{\otimes N})/N < \infty$ holds, for example, if we set the law of $\underline{x}^N(0) = (x_1^N(0), \ldots, x_N^N(0))$ to be $\mu_\circ^{\otimes N}$, where $\mu_\circ$ is any probability measure on $\mathbb{R}^d$ satisfying $\mathsf{KL}(\mu_\circ \| \pi) < \infty$.*

**Remark 2** (Optimality and dimension dependence)**.** *According to Sriperumbudur (2016); Hagrass et al. (2024), we have under mild regularity conditions, that the empirical measure $P_N := \frac{1}{N} \sum_{i=1}^N \delta_{X_i}$, where $X_i \sim P$, i.i.d., satisfies $\mathbb{E}[\mathsf{KSD}(P_N \| P)] = O(1/\sqrt{N})$. This points to the fact that our rates in Theorem 1 are presumably optimal with respect to $N$. While there is no curse-of-dimensionality in the $\mathsf{KSD}$ rates, the dimension factor appears in the numerator of the bound equation 8. When $\mathsf{k}(u, v) = \Psi(u - v)$ as in Remark 1 with $\sup_{x \in \mathbb{R}^d} \Delta V(x) \le Cd$ for some dimension independent constant $C$, it can be checked that $C^* \le \Psi(0)Cd - \Delta\Psi(0)$, which gives a linear in $d$ upper bound on $C^*$ for a wide range of kernels (including the Gaussian kernel) and potentials. Moreover, as long as mild regularity conditions are assumed about the kernel and the potential function, then according to Vempala & Wibisono (2019, Lemma 1), the initialization dependent term could be taken to be linear in $d$ when $V$ has Lipschitz gradients. These combine to give an $O(d/\sqrt{N})$ bound on the $\mathsf{KSD}$.*

*Proof of Theorem 1.* We will abbreviate $\mathsf{H}(t) := \mathsf{KL}(p^N(t) \| \pi^{\otimes N})$. Using the particle dynamics equation 2 and integration by parts, it is easy to verify that $p(t, \underline{z})$ is a weak solution of the following $N$-body Liouville equation (see, for example, Golse et al. (2013, Pg. 7) and Ambrosio et al. (2005, Chapter 8)) given by

$$\partial_t p(t, \underline{z}) + \frac{1}{N} \sum_{k,\ell=1}^N \mathrm{div}_{z_k}(p(t, \underline{z})\Phi(z_k, z_\ell)) = 0,$$

where $\Phi(z, w) := -\mathsf{k}(z, w)\nabla V(w) + \nabla_2 \mathsf{k}(z, w)$. Recalling equation 6, and using the density regularity obtained in Lemma 1, we have that

$$\begin{aligned}
\mathsf{H}'(t) &= \int \partial_t p(t, \underline{z})\mathrm{d}\underline{z} + \int \log\left(\frac{p(t, \underline{z})}{\pi^{\otimes N}(\underline{z})}\right) \partial_t p(t, \underline{z})\mathrm{d}\underline{z} \\
&= -\int \frac{1}{N} \sum_{k,\ell} \log\left(\frac{p(t, \underline{z})}{\pi^{\otimes N}(\underline{z})}\right) \mathrm{div}_{z_k}(p(t, \underline{z})\Phi(z_k, z_\ell))\mathrm{d}\underline{z} \\
&= \frac{1}{N} \sum_{k,\ell} \int \nabla_{z_k} \log\left(\frac{p(t, \underline{z})}{\pi^{\otimes N}(\underline{z})}\right) \cdot (p(t, \underline{z})\Phi(z_k, z_\ell))\mathrm{d}\underline{z} \\
&= \frac{1}{N} \sum_{k,\ell} \int \nabla_{z_k} p(t, \underline{z}) \cdot \Phi(z_k, z_\ell)\mathrm{d}\underline{z} + \frac{1}{N} \sum_{k,\ell} \int \nabla V(z_k) \cdot \Phi(z_k, z_\ell)p(t, \underline{z})\mathrm{d}\underline{z} \\
&= \frac{1}{N} \sum_{k,\ell} \int (-\mathrm{div}_{z_k} \Phi(z_k, z_\ell) + \nabla V(z_k) \cdot \Phi(z_k, z_\ell)) \, p(t, \underline{z})\mathrm{d}\underline{z}.
\end{aligned}$$

Now, observe that

$$\begin{aligned}
-\mathrm{div}_{z_k} \Phi(z_k, z_\ell) &= \mathrm{div}_{z_k}(\mathsf{k}(z_k, z_\ell)\nabla V(z_\ell)) - \mathrm{div}_{z_k}(\nabla_2 \mathsf{k}(z_k, z_\ell)) \\
&= \nabla_1 \mathsf{k}(z_k, z_\ell) \cdot \nabla V(z_\ell) - \nabla_1 \cdot \nabla_2 \mathsf{k}(z_k, z_\ell) + C^*(z_k)\mathbb{1}_{\{k=\ell\}}.
\end{aligned}$$

Similarly,

$$\nabla V(z_k) \cdot \Phi(z_k, z_\ell) = -\nabla V(z_k) \cdot (\mathsf{k}(z_k, z_\ell)\nabla V(z_\ell)) + \nabla V(z_k) \cdot \nabla_2 \mathsf{k}(z_k, z_\ell).$$

Therefore, using the explicit form of $\mathsf{KSD}$ in equation 5, we have

$$\sum_{k,\ell} (-\mathrm{div}_{z_k} \Phi(z_k, z_\ell) + \nabla V(z_k) \cdot \Phi(z_k, z_\ell)) = -N^2 \mathsf{KSD}^2(\mu(\underline{z}) \| \pi) + \sum_k C^*(z_k),$$

where $\mu(\underline{z}) := \frac{1}{N}\sum_{i=1}^{N}\delta_{z_i}$. Hence, we have

$$\mathsf{H}'(t) = -N\mathbb{E}[\mathsf{KSD}^2(\mu^N(t)\|\pi)] + \frac{1}{N}\mathbb{E}\Big[\sum_k C^*\,(x_k(t))\Big], \tag{9}$$

where we recall that $\mu^N(t) = \frac{1}{N}\sum_{i=1}^{N}\delta_{x_i(t)}$ is the empirical measure. Hence, we have

$$\frac{1}{T}\int_0^T \mathbb{E}[\mathsf{KSD}^2(\mu^N(t)\|\pi)]\mathrm{d}t \le \frac{\mathsf{H}(0)}{NT} + \frac{1}{N^2 T}\int_0^T \mathbb{E}\left[\sum_k C^*\,(x_k(t))\right]\mathrm{d}t,$$

which completes the first claim. The entropy bound follows from equation 9.

To prove the final claim, recall $\mu_{av}^N(\mathrm{d}x) := \frac{1}{N}\int_0^N \mu^N(t,\mathrm{d}x)\mathrm{d}t$ and note that the map $Q \mapsto \mathsf{KSD}(Q\|\pi)$ is convex, which follows immediately from the representation of KSD given in equation 4. From this and repeated applications of Jensen's inequality, we obtain

$$\mathbb{E}[\mathsf{KSD}(\mu_{av}^N\|\pi)] \le \frac{1}{N}\int_0^N \mathbb{E}[\mathsf{KSD}(\mu^N(t)\|\pi)]\mathrm{d}t \le \frac{1}{N}\int_0^N \sqrt{\mathbb{E}[\mathsf{KSD}^2(\mu^N(t)\|\pi)]}\mathrm{d}t$$

$$\le \left(\frac{1}{N}\int_0^N \mathbb{E}[\mathsf{KSD}^2(\mu^N(t)\|\pi)]\mathrm{d}t\right)^{1/2} \le \frac{\left(\sup_L \frac{\mathsf{H}(0)}{L} + C^*\right)^{\frac{1}{2}}}{\sqrt{N}}.$$

This completes the proof of the theorem. $\qquad\square$

We now address the convergence in law of the time-averaged marginals of a single particle when the initial particle locations are drawn from an exchangeable law. We defer its proof to Appendix A.2.

**Theorem 2.** *Suppose Assumption 1 holds, $C^* < \infty$ and let $k(u,v) = \Psi(x-y)$, where $\Psi$ is a $C^3$ function with non-vanishing generalized Fourier transform. Suppose also that the law $p_0$ of the initial particle locations $(x_1(0),\ldots,x_N(0))$ is exchangeable for each $N \in \mathbb{N}$ and $\limsup_{N\to\infty}\frac{1}{N}\mathsf{KL}(p^N(0)\|\pi^{\otimes N}) < \infty$. Define*

$$\bar{\mu}^N(A) := \frac{1}{N}\int_0^N \mathbb{P}(x_1(t) \in A)\mathrm{d}t, \ \text{for} \ A \in \mathcal{B}(\mathbb{R}^d).$$

*Then, $\bar{\mu}^N \to \pi$, weakly.*

## 3 DISCRETE-TIME FINITE-PARTICLE RATES IN KSD METRIC

In this section, we obtain KSD rates for the discrete-time SVGD dynamics given by equation 1. The dynamics can be succinctly represented as

$$\underline{x}(n+1) = \underline{x}(n) - \eta\mathbf{T}(\underline{x}(n)), \ n \in \mathbb{N}_0,$$

where $\eta$ is the step-size and $\mathbf{T} = (\mathsf{T}_1,\ldots,\mathsf{T}_N)'$ with $\mathsf{T}_i(\underline{x}) = \frac{1}{N}\sum_j [k(x_i,x_j)\nabla V(x_j) - \nabla_2 k(x_i,x_j)]$. In this section, we use $T$ to denote the number of iterations.

Although the idea is once again to track the evolution of the relative entropy of the joint density of particle locations with respect to $\pi^{\otimes N}$, one needs a careful quantification of the discretization error to obtain an equation similar to equation 9. We will make the following assumptions. All constants appearing in the section will be independent of $d, N$.

**Assumption 2.** *We make the following regularity assumptions.*

    *(a)* Boundedness*: $k$ and all its partial derivatives up to order $2$ are uniformly bounded by $B \in (0,\infty)$.*

    *(b)* Growth*: $\inf_{z\in\mathbb{R}^d} V(z) > 0$ and, for some $A > 0$, $\alpha \in [0,1/2]$, $\|\nabla V(x)\| \le AV(x)^\alpha$ for all $x \in \mathbb{R}^d$.*

(c) *Bounded Hessian:* $\sup_{z \in \mathbb{R}^d} \|H_V(z)\|_{op} = C_V < \infty$, *where $H_V$ denotes the Hessian of $V$ and $\| \cdot \|_{op}$ denotes the operator norm.*

(d) *Recalling equation 7, we assume that $\sup_{z \in \mathbb{R}^d} \nabla_2 k(z,z) \cdot \nabla V(z) + k(z,z)\Delta V(z) - \Delta_2 k(z,z) = c^* d$ for some constant $c^* > 0$.*

(e) *Initial entropy bound:* $\mathsf{KL}(p(0)||\pi^{\otimes N}) \leq C_{KL} N d$ *for some constant $C_{KL} > 0$.*

**Remark 3.** *Conditions (a) and (c) are essentially motivated by the work of Korba et al. (2020), who in turn are motivated by standard assumptions made in the stochastic optimization literature. The condition (b) in Assumption 2 captures the growth rate of the potential $V$ which plays a crucial role in the step-size selection and number of iterations required (see equation 10 below) to obtain the convergence rate in Theorem 3. As $\alpha$ approaches $0$, $\pi$ approaches the exponential distribution and as $\alpha$ approaches $1/2$, $\pi$ is close to a Gaussian (in tail behavior). Conditions (d) and (e) are explicit refinements of similar conditions required in the continuous-time analysis.*

**Theorem 3.** *Suppose Assumption 2 holds. Let the initial locations $(x_1(0), \ldots, x_N(0))$ be sampled from*

$$p_K(0) := p(0)|\mathcal{S}_K \quad where \quad \mathcal{S}_K := \Big\{ \underline{x} \in \left(\mathbb{R}^d\right)^N : N^{-1} \sum_i V(x_i) \leq K \Big\},$$

*for some $K > 0$ satisfying $\int_{\mathcal{S}_K} p(0, \underline{z}) d\underline{z} \geq 1/2$, where $p(0)|\mathcal{S}_K$ represents the restriction of the density to the set $\mathcal{S}_K$. Then there exist positive constants $a, b$ depending only on the constants appearing in Assumption 2 such that with*

$$\eta = \frac{a}{d^{\frac{1+\alpha}{2(1-\alpha)}} + \sqrt{d}K^\alpha + d} N^{-\frac{1+\alpha}{1-\alpha}}, \qquad T = \lceil N^{\frac{2}{1-\alpha}} \rceil, \tag{10}$$

*we have*

$$\mathbb{E}_{p_K(0)} \Big[ \frac{1}{T} \sum_{n=0}^{T-1} \mathsf{KSD}^2 \left( \mu_n^N || \pi \right) \Big] \leq \frac{bd \left( d^{\frac{1+\alpha}{2(1-\alpha)}} + \sqrt{d}K^\alpha + d \right)}{N}.$$

*In particular, if $(x_1(0), \ldots, x_N(0))$ is sampled from $p(0)$, then for any $\epsilon > 0$,*

$$\mathbb{P}\left( \frac{1}{T} \sum_{n=0}^{T-1} \mathsf{KSD}^2 \left( \mu_n^N || \pi \right) > \frac{bd \left( d^{\frac{1+\alpha}{2(1-\alpha)}} + \sqrt{d}K^\alpha + d \right)}{N\epsilon} \right) \leq \epsilon + \int_{\mathcal{S}_K^c} p(0, \underline{z}) d\underline{z}.$$

We prove Theorem 3 in Appendix A.3.

**Remark 4.** *The proof of Theorem 3 involves an 'interpolation' in the spirit of Korba et al. (2020) between the laws of $\underline{x}(n)$ and $\underline{x}(n+1)$ which leads to a Taylor expansion of the relative entropy. The main subtlety in the analysis when dealing with the joint law evolution here comes from the fact that, unlike for the population limit SVGD, the Hilbert-Schmidt norm of the Jacobian of the map $\mathbf{T}(\underline{x})$ at $\underline{x} = \underline{x}(n)$ is not uniformly bounded in $n$ and $\underline{x}(0)$. This requires refined 'a priori' bounds on the rate of growth of the path functional $n \mapsto \frac{1}{N} \sum_i V(x_i(n))$ (see Lemma 3). As a result, for initial locations in $\mathcal{S}_K$ one can fine tune the step-size depending on $N, d, K$ such that the second order term in the Taylor expansion becomes small in comparison to the first term. Moreover, the first order term is shown to have the same form as the continuous-time derivative of the joint relative entropy obtained in the proof of Theorem 1. This leads to the results in Theorem 3.*

*Marginal convergence results, similar to Theorem 2, in the discrete-time setting can also be deduced from the entropy bounds involved in proving Theorem 3.*

## 4 CONTINUOUS-TIME FINITE-PARTICLE RATES IN $W_2$ METRIC

We now explore convergence rates for the continuous-time SVGD in the $L^2$-Wasserstein metric. For $s > 0$, let $\mathcal{P}_s$ be the set of all Borel measurable probability measures on $\mathbb{R}^d$ with finite $s$-moment. For two measures $\mu, \nu \in \mathcal{P}_s$, the $L^s$-Wasserstein distance (based on the Euclidean distance) is defined as

$$\mathsf{W}_s(\mu, \nu) := \left( \inf_{\pi \in \Pi(\mu, \nu)} \mathbb{E}_{(X,Y) \sim \pi} \left[ \|X - Y\|_2^s \right] \right)^{1/s},$$

where $\Pi(\mu, \nu)$ denotes the set of all possible couplings of the probability measures $\mu$ and $\nu$.

To address $W_2$ convergence, we consider the SVGD dynamics in equation 2 with kernels of the form

$$\tilde{k}(u, v) = 1 + \langle u, v \rangle + \Psi(u - v). \tag{11}$$

We further assume that the kernel obtained by $(u, v) \mapsto \Psi(u - v)$ is positive-definite, $\Psi(z) = \Psi(-z)$ for all $z \in \mathbb{R}^d$, $\sup_{z \in \mathbb{R}^d} \|\nabla\Psi(z)\| < \infty$, $\Psi \in \mathcal{C}^3$ and has a non-vanishing and continuous generalized Fourier transform. A classical example of a kernel that satisfies these assumptions is the Matérn kernel. We first state the following dissipativity and growth assumptions from Kanagawa et al. (2022) along with an additional Laplacian growth condition.

**Assumption 3.** *The following conditions hold:*

    *(a) Dissipativity: The potential $V$ satisfies $-\langle x, \nabla V(x) \rangle \leq -\alpha\|x\|^2 + \beta_1\|x\| + (\beta_0 - d)$, for some $\alpha > 0$ and $\beta_1, \beta_0 \geq 0$.*

    *(b) Growth: $\|\nabla V(x)\| \leq \lambda_b(1 + \|x\|)$, for some $\lambda_b > 0$.*

    *(c) $\sup_{z \in \mathbb{R}^d} \Delta V(z) < \infty$.*

The above assumptions are widely used in the MCMC literature (e.g., Raginsky et al. (2017)) and are satisfied in many cases including certain Gaussian mixture models, and allow for some degree of non-log-concavity in $\pi$.

**Theorem 4.** *Let Assumption 3 hold. Assume that the initialization is such that $\limsup_{N \to \infty} N^{-1}\mathsf{KL}(p(0)\|\pi^{\otimes N}) < \infty$. Fix any $\sigma > 0$ and let $M = M(N) := \lceil N^{2+\sigma} \rceil$. Then, there exists a constant $C_0 > 0$ such that*

$$\mathbb{E}[\mathsf{KSD}(\mu_{av}^M\|\pi)] \leq \frac{C_0}{N^{1+\sigma/2}}, \quad \forall N \geq 1 \qquad and \qquad W_2(\mu_{av}^M, \pi) \xrightarrow{a.s} 0, \ as \ N \to \infty,$$

*where recall $\mu_{av}^M(\mathrm{d}x) := \frac{1}{M}\int_0^M \mu^M(t, \mathrm{d}x)\mathrm{d}t$.*

Theorem 4 is proven in Appendix A.4. By the results in Kanagawa et al. (2022, Section 3.2), we can translate the KSD bound in Theorem 4 into a bound in the $W_2$ metric. While a more abstract result for the choice of kernel in equation 11 could be obtained by leveraging Kanagawa et al. (2022, Theorem 3.2), for the sake of concreteness, we restrict ourselves to the Matérn-family of kernels. Specifically we consider equation 11 with

$$\tilde{k}_{mk}(u, v) := 1 + \langle u, v \rangle + \underbrace{\frac{2^{1-(d/2+\nu)}}{\Gamma(d/2+\nu)}\|\Sigma(u-v)\|_2^\nu K_{-\nu}\big(\|\Sigma(u-v)\|_2\big)}_{:=\Psi_{mk}(u-v)}, \tag{12}$$

where $\Gamma$ is the Gamma function, $\Sigma$ a strictly positive definite matrix, and $K_{-\nu}$ the modified Bessel function of the second kind of order $-\nu$.

To proceed, we also require the following assumption from Kanagawa et al. (2022), which is motivated by the Langevin diffusion:

$$\mathrm{d}Z_t = -\nabla V(Z_t)\mathrm{d}t + \sqrt{2}\mathrm{d}B_t, \tag{13}$$

where $(B_t)_{t \geq 0}$ is a $d$-dimensional Brownian motion. Note that equation 13 is the stochastic differential equation equivalent of the Wasserstein gradient flow in equation 3; see, for example, Jordan et al. (1998); Bakry et al. (2014) for details. The connection between the two perspectives has in particular proved to be extremely useful for analyzing both Markov Chain Monte Carlo algorithms and particle-based methods.

**Assumption 4.** *For $s \in \{1, 2\}$, let $\rho_s : [0, \infty) \to [0, \infty)$ be an upper bounding function for the $L^s$-Wasserstein distance in the following sense:*

$$W_s(\mathcal{L}(Z_t^x), \mathcal{L}(Z_t^y)) \leq \rho_s(t)\|x - y\|_2, \ \forall x, y \in \mathbb{R}^d, t \geq 0, \tag{14}$$

*where $Z_t^x$ and $Z_t^x$ are Langevin diffusion processes in equation 13 with initializations $Z_0 = x$ and $Z_0 = y$. Assume that $\tilde{\rho}(t) := \frac{\log\frac{\rho_1(t)}{\rho_1(0)} - \log\rho_2(t)}{\log\frac{\rho_1(t)}{\rho_1(0)}}$ is uniformly bounded in $t$ and, moreover,*

$$\int_0^\infty \rho_1(t)\left\{1 + \sqrt{\rho_1(t)}\,\tilde{\rho}(t)\right\}\mathrm{d}t < \infty.$$

**Remark 5.** *Suppose there exist $U > 0$ and $R, L \geq 0$ such that the potential $V$ satisfies*

$$\frac{\langle \nabla V(x) - \nabla V(y), x - y \rangle}{\|x - y\|_2^2} \leq \begin{cases} -U & \text{if } \|x - y\|_2 > R, \\ L & \text{if } \|x - y\|_2 \leq R. \end{cases} \tag{15}$$

*Then, by Eberle (2011; 2016), there exists $c, c_1 > 0$ such that we can set $\rho_1(t) = ce^{-c_1 t}$, $t \geq 0$. Moreover, using equation 15 and Grönwall's lemma, we can set $\rho_2(t) = e^{Lt}$, $t \geq 0$. Consequently, $\tilde{\rho}(t) = \frac{L+c_1}{c_1}$ and hence Assumption 4 holds.*

**Theorem 5.** *Consider the SVGD updates in equation 2 with the kernel in equation 12. Suppose Assumption 4 and the assumptions made in Theorem 4 are satisfied. Then, with $\sigma, M$ as in Theorem 4, there exists a constant $C(d) > 0$ such that for any $\epsilon \in (0, 1)$, we have*

$$\mathbb{P}\left[ \mathsf{W}_2(\mu_{av}^M, \pi) \geq C(d) \left( \frac{C_0}{\epsilon N^{1+\sigma/2}} \right)^{r(d)} \right] \leq \epsilon, \qquad \forall N \geq \left( \frac{C_0}{\epsilon} \right)^{\frac{2}{2+\sigma}},$$

*where $C_0$ is the same constant from Theorem 4, $C(d)$ is the constant $C_{P,d}(1)$ from Kanagawa et al. (2022, Theorem 3.5), and*

$$r(d) := \frac{1}{3(\frac{4d+1}{d})} \cdot \frac{1}{\frac{3d}{2} + \frac{17}{6} + \left[ \frac{d+1}{d} + \frac{5}{3} \right] \nu}. \tag{16}$$

We prove in Theorem 5 in Appendix A.5.

**Remark 6.** *Note that $r(d) \approx \frac{1}{18d}$ for large $d$. Hence, unlike the KSD rates in Theorem 5, the $\mathsf{W}_2$ rates have a curse-of-dimensionality. However, this is expected, as even in the case of i.i.d. samples, we have a similar curse-of-dimensionality (Dudley, 1969; Weed & Bach, 2019). Intuitively, this can be understood by observing that convergence in KSD captures convergence of expectations for a class of test functions that is much smaller than that for Wasserstein convergence (see Gorham & Mackey (2017)). The latter class is large enough to be highly sensitive to the effect of growing dimension, thereby exhibiting the curse-of-dimensionality.*

## 5 PROPAGATION OF CHAOS

We now exhibit a *long-time propagation of chaos* (POC) for the particle system started from an exchangeable initial configuration and driven by the dynamics equation 2. More precisely, we show in the following proposition that, under the conditions of Theorem 4, the time-averaged marginals of particle locations over the time interval $[0, N]$ become asymptotically independent, with distribution $\pi$, as $N \to \infty$. This result shows in particular that, unlike traditional MCMC schemes (Brooks et al., 2011), the SVGD algorithm provides multiple i.i.d. approximate samples from the target distribution $\pi$. We defer its proof to Appendix A.6.

**Proposition 1.** *Suppose that the law $p_0^N$ of the initial particle locations $(x_1^N(0), \ldots, x_N^N(0))$ is exchangeable for each $N \in \mathbb{N}$. For $1 \leq k \leq N$, define the $k$-dimensional marginal of the time-averaged occupancy measure of particle locations as follows:*

$$\bar{\mu}_k^N(A_1, \ldots, A_k) := \frac{1}{N} \int_0^N \mathbb{P}(x_1^N(t) \in A_1, \ldots, x_k^N(t) \in A_k) \mathsf{d}t, \ \text{for} \ A_1, \ldots, A_k \in \mathcal{B}(\mathbb{R}^d).$$

*Recall $M = M(N) := \lceil N^{2+\eta} \rceil$. Under the same setting as Theorem 4, we have that, for any fixed $k \in \mathbb{N}$, $\mathsf{W}_1(\bar{\mu}_k^M, \pi^{\otimes k}) \overset{a.s}{\to} 0$, as $N \to \infty$.*

**Remark 7.** *This result should be compared with Shi & Mackey (2024, Theorem 2) and Lu et al. (2019, Proposition 2.6) where finite-time POC results are shown: the particle marginal laws at a fixed time become asymptotically independent as $N \to \infty$. This follows upon observing that the convergence of empirical measures shown in these papers implies a finite-time POC (Chaintron & Diez, 2022, Proposition 9 and Theorem 3.21). However, owing to the lack of Lipschitz property of the vector field driving equation 2, this POC can only be extended to growing times $t = t_N = O(\log \log N)$ when the particle marginal laws are not necessarily close to $\pi$. In contrast, Proposition 1 extends to the time interval $[0, N]$ (hence, long-time POC) and the time-averaged particle trajectories essentially produce i.i.d. samples from $\pi$.*

ACKNOWLEDGEMENTS

KB was supported in part by National Science Foundation (NSF) grant DMS-2413426. SB was supported in part by NSF CAREER award DMS-2141621 and NSF RTG grant DMS- 2134107. We thank Lester Mackey for his insightful comments on an earlier version of this work and for valuable discussions regarding the broader literature on KSD and SVGD.

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

## A  PROOFS

### A.1  PROOF OF LEMMA 1

Let $F : \left(\mathbb{R}^d\right)^N \to \left(\mathbb{R}^d\right)^N$ be given by

$$F_i(\underline{z}) := -\frac{1}{N} \sum_j \mathsf{k}(z_i, z_j) \nabla V(z_j) + \frac{1}{N} \sum_j \nabla_2 \mathsf{k}(z_i, z_j),$$

for $1 \le i \le N$. By Assumption 1, $F$ is a $\mathcal{C}^2$ map. Note that the SVGD particle trajectories can be written as $\{\mathbf{x}(t, \underline{x}(0)) : t \ge 0\}$, where the flow $\mathbf{x} : [0, \infty) \times \left(\mathbb{R}^d\right)^N \to \left(\mathbb{R}^d\right)^N$ is given by

$$\dot{\mathbf{x}}(t, \underline{z}) = F(\mathbf{x}(t, \underline{z})), \quad \mathbf{x}(0, \underline{z}) = \underline{z},$$

with $\dot{\mathbf{x}}$ denoting the time derivative. By Hartman (2002, Chapter 5, Cor. 4.1), the map $(t, \underline{z}) \mapsto \mathbf{x}(t, \underline{z})$, and consequently, the inverse map $(t, \underline{z}) \mapsto \mathbf{x}(t, \cdot)^{-1}(\underline{z})$, are $\mathcal{C}^2$ maps on $(0, \infty) \times \left(\mathbb{R}^d\right)^N$.

A simple change of variable formula gives (see Crippa (2008, Page 21))

$$p(t, \underline{z}) = \frac{p_0}{\det(\nabla \mathbf{x}(t, \cdot))} \circ \mathbf{x}(t, \cdot)^{-1}(\underline{z}), \quad (t, \underline{z}) \in (0, \infty) \times \left(\mathbb{R}^d\right)^N.$$

The existence and regularity of $p(\cdot, \cdot)$ then follows from the above observations.

### A.2  PROOF OF THEOREM 2

We will first show tightness of $\{\bar{\mu}^N\}_N$. By subadditivity of relative entropy (which follows from Budhiraja & Dupuis (2019, Lemma 2.4(b) and Theorem 2.6)), we have

$$\mathsf{KL}(\mathcal{L}(x_1(t))||\pi) \le \frac{1}{N}\mathsf{KL}(p(t)||\pi^{\otimes N}) \le \frac{\mathsf{KL}(p(0)||\pi^{\otimes N})}{N} + \frac{C^* t}{N}.$$

Hence, there exists $C > 0$ such that $\mathsf{KL}(\mathcal{L}(x_1(t))||\pi) \le C$ for all $t \in [0, N]$, $\forall N \ge 1$. Fix any $\epsilon > 0$. Let $\delta > 0$ such that $\delta C < \epsilon/2$. Let $K$ be a compact subset of $\mathbb{R}^d$ such that $\delta \log(1 + \pi(K^c)(e^{1/\delta} - 1)) \le \epsilon/2$. By the variational representation of relative entropy (Budhiraja & Dupuis, 2019, Prop. 2.3) and Theorem 1, we have

$$\mathbb{P}(x_1(t) \notin K) \le \delta \left[ \log\left( \int e^{\frac{1}{\delta} \mathbb{1}_{K^c}(z)} \pi(\mathsf{d}z) \right) + \mathsf{KL}(\mathcal{L}(x_1(t))||\pi) \right]$$

$$\le \delta \log(1 + \pi(K^c)(e^{1/\delta} - 1)) + \delta C < \epsilon$$

for all $t \in [0, N]$, $\forall N \geq 1$. In particular, $\{\bar{\mu}^N\}_N$ is tight. Moreover, we have

$$\mathsf{KSD}(\bar{\mu}^N \| \pi) = \mathsf{KSD}\left(\frac{1}{N} \int_0^N \mathcal{L}(x_1(t))\mathrm{d}t \| \pi\right)$$

$$\leq \frac{1}{N} \int_0^N \mathsf{KSD}(\mathcal{L}(x_1(t)) \| \pi)\mathrm{d}t \qquad \textit{by the convexity of } \mathsf{KSD}$$

$$= \frac{1}{N} \int_0^N \mathsf{KSD}(\mathbb{E}[\mu^N(t)] \| \pi) \quad \textit{using exchangeability, where } \mathbb{E}[\mu^N(t)](\mathrm{d}x) := \mathbb{E}[\mu^N(t, \mathrm{d}x)]$$

$$\leq \frac{1}{N} \int_0^N \mathbb{E}[\mathsf{KSD}(\mu^N(t) \| \pi)]\mathrm{d}t \to 0 \qquad \textit{by } \mathsf{KSD} \textit{ convexity and Theorem 1}.$$

The result now follows from Gorham & Mackey (2017, Theorem 7).

### A.3 PROOF OF THEOREM 3

The proof of Theorem 3 proceeds through the following lemmas. We will assume throughout that Assumption 2 holds.

**Lemma 2.** $\|\mathbf{T}(\underline{x})\|^2 \leq 2A^2B^2N \left(\frac{1}{N}\sum_i V(x_i)\right)^{2\alpha} + 2NB^2d, \quad \underline{x} \in \left(\mathbb{R}^d\right)^N.$

*Proof of Lemma 2.* Observe that

$$\|\mathbf{T}(\underline{x})\|^2 = \frac{1}{N^2}\sum_i \|\sum_j (\mathsf{k}(x_i, x_j)\nabla V(x_j) - \nabla_2 \mathsf{k}(x_i, x_j)\|^2$$

$$\leq \frac{2}{N}\sum_{i,j}\left(\mathsf{k}^2(x_i, x_j)\|\nabla V(x_j)\|^2 + \|\nabla_2 \mathsf{k}(x_i, x_j)\|^2\right)$$

$$\leq 2A^2B^2N\left(\frac{1}{N}\sum_j V(x_j)^{2\alpha}\right) + 2NB^2d$$

$$\leq 2A^2B^2N\left(\frac{1}{N}\sum_j V(x_j)\right)^{2\alpha} + 2NB^2d,$$

where the last step follows by Jensen's inequality noting $\alpha \leq 1/2$. $\qquad\square$

The following lemma gives a key 'a priori' bound on the growth rate of $N^{-1}\sum_i V(x_i(n))$ in terms of $n$ and $\eta$ and is crucial to the rest of the proof.

**Lemma 3.** *There exist positive constants $M, D$ depending only on the constants appearing in Assumption 2 such that for $T \geq 1$, $0 \leq n \leq T$, $\eta \leq 1 \wedge \frac{1}{\sqrt{DT}}$,*

$$\frac{1}{N}\sum_i V(x_i(n)) \leq M\left(d^{\frac{1}{1-\alpha}} + \frac{1}{N}\sum_i V(x_i(0))\right)\left[(\eta n)^{\frac{1}{1-\alpha}} \vee 1\right].$$

*Proof of Lemma 3.* Note that, using Taylor's theorem,

$$\frac{1}{N}\sum_i V(x_i(n+1)) - \frac{1}{N}\sum_i V(x_i(n)) = \frac{1}{N}\sum_i \langle \nabla V(x_i(n)), x_i(n+1) - x_i(n)\rangle$$

$$+ \frac{1}{N}\sum_i \int_0^1 (1-s)\langle x_i(n+1) - x_i(n), H_V(x_i(n) - s\eta\mathsf{T}_i(\underline{x}(n)))(x_i(n+1) - x_i(n))\rangle ds$$

$$\leq \frac{1}{N}\sum_i \langle \nabla V(x_i(n)), x_i(n+1) - x_i(n)\rangle + \frac{C_V\eta^2}{2N}\|\mathbf{T}(\underline{x}(n))\|^2.$$

Now, applying Lemma 2 in the above and writing $D_1 = A^2 B^2 C_V$ and $D_2 = B^2 C_V$, we obtain

$$\frac{1}{N} \sum_i V(x_i(n+1)) - \frac{1}{N} \sum_i V(x_i(n))$$

$$\leq \frac{1}{N} \sum_i \langle \nabla V(x_i(n)), x_i(n+1) - x_i(n) \rangle + D_1 \eta^2 \left( \frac{1}{N} \sum_j V(x_j(n)) \right)^{2\alpha} + D_2 d\eta^2$$

$$= -\frac{\eta}{N^2} \sum_{i,j} \langle \nabla V(x_i(n)), \mathsf{k}(x_i(n), x_j(n)) \nabla V(x_j(n)) \rangle + \frac{\eta}{N^2} \sum_{i,j} \langle \nabla V(x_i(n)), \nabla_2 \mathsf{k}(x_i(n), x_j(n)) \rangle$$

$$+ D_1 \eta^2 \left( \frac{1}{N} \sum_j V(x_j(n)) \right)^{2\alpha} + D_2 d\eta^2$$

$$\leq \frac{\eta}{N^2} \sum_{i,j} \langle \nabla V(x_i(n)), \nabla_2 \mathsf{k}(x_i(n), x_j(n)) \rangle + D_1 \eta^2 \left( \frac{1}{N} \sum_j V(x_j(n)) \right)^{2\alpha} + D_2 d\eta^2,$$

where, in the last step, we used the positive-definiteness of $\mathsf{k}$. Thus, suppressing the dependence on $n$ of the right hand side to avoid cumbersome notation, we obtain

$$\frac{1}{N} \sum_i V(x_i(n+1)) - \frac{1}{N} \sum_i V(x_i(n))$$

$$\leq \frac{B\sqrt{d}\eta}{N} \sum_i \|\nabla V(x_i)\| + D_1 \eta^2 \left( \frac{1}{N} \sum_j V(x_j) \right)^{2\alpha} + D_2 d\eta^2$$

$$\leq AB\sqrt{d}\eta \left( \frac{1}{N} \sum_i V(x_i) \right)^{\alpha} + D_1 \eta^2 \left( \frac{1}{N} \sum_j V(x_j) \right)^{2\alpha} + D_2 d\eta^2.$$

Thus, writing $f(n) := \frac{1}{N} \sum_i V(x_i(n))$, recalling $\alpha \leq 1/2$ and using $f(n)^{2\alpha} \leq 1 + f(n)$, we obtain

$$f(n+1) \leq (1 + D_1 \eta^2) f(n) + AB\sqrt{d}\eta f(n)^{\alpha} + D_3 d\eta^2, \tag{17}$$

where $D_3 = D_1 + D_2$. We will now harness the recursive bound equation 17 to obtain the claimed bound in the lemma. First, we handle the case $0 \leq n \leq \lceil 1/\eta \rceil$. Fix $L > 0$. Define

$$\tau_L := \sup\{n \geq 0 : f(n) \leq L\} \wedge \lceil 1/\eta \rceil.$$

Then for $1 \leq n \leq \tau_L$, equation 17 gives for $\eta \leq 1$,

$$f(n) \leq (1 + D_1 \eta^2) f(n-1) + AB\sqrt{d}\eta L^{\alpha} + D_3 d\eta^2$$

$$\leq \sum_{\ell=0}^{n-1} (1 + D_1 \eta^2)^{\ell} (AB\sqrt{d}\eta L^{\alpha} + D_3 d\eta^2) + (1 + D_1 \eta^2)^n f(0)$$

$$\leq \frac{2}{\eta} (1 + D_1 \eta^2)^{1/\eta} (AB\sqrt{d}\eta L^{\alpha} + D_3 d\eta^2) + (1 + D_1 \eta^2)^{1+1/\eta} f(0)$$

$$\leq 2e^{D_1} (AB\sqrt{d} L^{\alpha} + D_3 d) + (1 + D_1) e^{D_1} f(0).$$

Hence, taking $L = \hat{M} (d + f(0))$ for some suitably large $\hat{M} \geq 1$ depending only on the constants appearing in Assumption 2, we conclude from the above bound that $f(\tau_L) < L$ and hence $\tau_L = \lceil 1/\eta \rceil$. Thus,

$$f(n) \leq \hat{M} (d + f(0)) \quad \text{for all } 0 \leq n \leq \lceil 1/\eta \rceil.$$

Now, we handle the case $1/\eta \leq n \leq T$. We will proceed by induction. Let

$$\beta := M(d^{\frac{1}{1-\alpha}} + f(0)), \quad \text{where} \quad M = \hat{M} \vee [16(1-\alpha)^2 (AB + D_3)]^{\frac{1}{1-\alpha}}.$$

Take any $\eta \leq 1 \wedge \frac{1}{4(1-\alpha)\sqrt{D_1 T}}$. Then, by the above bound, note that for $n = \lceil 1/\eta \rceil$, $f(n) \leq \beta(n\eta)^{\frac{1}{1-\alpha}}$. Suppose for some $1/\eta \leq n \leq T$, $f(n) \leq \beta(n\eta)^{\frac{1}{1-\alpha}}$. Then, by equation 17,

$$
\begin{aligned}
f(n+1) &\leq (1 + D_1\eta^2)\beta(n\eta)^{\frac{1}{1-\alpha}} + AB\sqrt{d}\eta\beta^\alpha(n\eta)^{\frac{\alpha}{1-\alpha}} + D_3 d\eta^2 \\
&\leq \beta((n+1)\eta)^{\frac{1}{1-\alpha}} \left[ (1 + D_1\eta^2)\left(\frac{n}{n+1}\right)^{\frac{1}{1-\alpha}} + \frac{AB\sqrt{d}}{\beta^{1-\alpha}n} + \frac{D_3 d\eta^{\frac{1-2\alpha}{1-\alpha}}}{\beta n^{\frac{1}{1-\alpha}}} \right] \\
&\leq \beta((n+1)\eta)^{\frac{1}{1-\alpha}} \left[ (1 + D_1\eta^2) - \left\{ \frac{1}{8(1-\alpha)^2 n} - \frac{AB\sqrt{d} + D_3 d}{\beta^{1-\alpha}n} \right\} \right] \\
&\leq \beta((n+1)\eta)^{\frac{1}{1-\alpha}} \left[ (1 + D_1\eta^2) - \left\{ \frac{1}{8(1-\alpha)^2 n} - \frac{AB\sqrt{d} + D_3 d}{M^{1-\alpha}dn} \right\} \right] \\
&\leq \beta((n+1)\eta)^{\frac{1}{1-\alpha}} \left[ 1 + D_1\eta^2 - \frac{1}{16(1-\alpha)^2 n} \right] \\
&\leq \beta((n+1)\eta)^{\frac{1}{1-\alpha}},
\end{aligned}
$$

where the fifth inequality uses the choice of $M$ and the last inequality uses the bound on $\eta$. The claimed bound follows by induction. □

We will now use Lemma 3 to obtain bounds on $\|\mathbf{T}(\underline{x}(n))\|^2$ and the Hilbert-Schmidt norm of the Jacobian matrix $J\mathbf{T}(\underline{x}(n))$.

**Lemma 4.** *For $T \geq 1$, $0 \leq n \leq T$, $\eta \leq 1 \wedge \frac{1}{\sqrt{DT}}$, we have*

$$
\|\mathbf{T}(\underline{x}(n))\|^2 \leq 2A^2 B^2 N \left[ M^{2\alpha}\left( d^{\frac{1}{1-\alpha}} + \frac{1}{N}\sum_i V(x_i(0)) \right)^{2\alpha} \left[ (\eta n)^{\frac{2\alpha}{1-\alpha}} \vee 1 \right] \right] + 2NB^2 d,
$$

$$
\begin{aligned}
\|J\mathbf{T}(\underline{x}(n))\|_{HS}^2 &\leq 8A^2 B^2 d(N+2) \left[ M^{2\alpha}\left( d^{\frac{1}{1-\alpha}} + \frac{1}{N}\sum_i V(x_i(0)) \right)^{2\alpha} \left[ (\eta n)^{\frac{2\alpha}{1-\alpha}} \vee 1 \right] \right] \\
&\quad + 8B^2 d^2(N+3) + 8B^2 dC_V^2.
\end{aligned}
$$

*Proof of Lemma 4.* The first bound follows from Lemma 2 and Lemma 3. To prove the second bound, note that

$$
\|J\mathbf{T}(\underline{x})\|_{HS}^2 = \sum_{i,j=1}^N \sum_{k,l=1}^d \|\partial_{jl}\mathsf{T}_{ik}(\underline{x})\|^2,
$$

where for $v_i \in \mathbb{R}^d$, $v_{ik}$ denotes the $k$th coordinate of $v_i$ and $\partial_{jl}$ denotes the partial derivative with respect to $x_{jl}$. We will also write for $m = 1, 2$, $\partial_{mk}\mathsf{k}$ to denote the partial derivative of $\mathsf{k}$ with respect to the $k$th coordinate of the $m$th variable. Observe that,

$$
\begin{aligned}
\partial_{x_{jl}}\mathsf{T}_{ik}(\underline{x}) &= \frac{1}{N}\sum_{u=1}^N \left( \partial_{1l}\mathsf{k}(x_i, x_u)\partial_k V(x_u) - \partial_{2k}\mathsf{k}(x_i, x_u) \right) \mathbb{1}(i = j) \\
&\quad + \frac{1}{N}\left( \partial_{2l}\mathsf{k}(x_i, x_j)\partial_k V(x_j) - \partial_{2k}\mathsf{k}(x_i, x_j) \right) \\
&\quad + \frac{1}{N}\left( \mathsf{k}(x_i, x_j)\partial_{lk}V(x_j) - \partial_{2k}\mathsf{k}(x_i, x_j) \right) \\
&\quad + \frac{1}{N}\left( \mathsf{k}(x_i, x_j)\partial_k V(x_j) - \partial_{2l}\partial_{2k}\mathsf{k}(x_i, x_j) \right).
\end{aligned}
$$

Write $S_m(i,j)$ for the $m$th term in the above bound for $m = 1, 2, 3, 4$. In a similar manner as the proof of Lemma 2,

$$
\begin{aligned}
\sum_{ijkl} |S_1(i,j)|^2 &\leq \frac{1}{N} \sum_{iukl} \left( \partial_{1l}\mathsf{k}(x_i, x_u)\partial_k V(x_u) - \partial_{2k}\mathsf{k}(x_i, x_u) \right)^2 \\
&= \frac{1}{N} \sum_{iul} \left\| \partial_{1l}\mathsf{k}(x_i, x_u)\nabla V(x_u) - \nabla_2\mathsf{k}(x_i, x_u) \right\|^2 \\
&\leq 2A^2 B^2 dN \left( \frac{1}{N}\sum_i V(x_i) \right)^{2\alpha} + 2NB^2 d^2.
\end{aligned}
$$

Similarly,

$$
\begin{aligned}
\sum_{ijkl} |S_2(i,j)|^2 &= \frac{1}{N^2} \sum_{ijkl} \left( \partial_{2l}\mathsf{k}(x_i, x_j)\partial_k V(x_j) - \partial_{2k}\mathsf{k}(x_i, x_j) \right)^2 \\
&\leq \frac{2B^2}{N^2} \sum_{ij} (d\|\nabla V(x_j)\|^2 + d^2) \\
&\leq 2A^2 B^2 d \left( \frac{1}{N}\sum_i V(x_i) \right)^{2\alpha} + 2B^2 d^2.
\end{aligned}
$$

Moreover,

$$
\begin{aligned}
\sum_{ijkl} |S_3(i,j)|^2 &\leq \frac{2B^2}{N^2} \sum_{ijkl} \left( (\partial_{lk} V(x_j))^2 + 1 \right) \\
&\leq 2B^2 \sup_{z\in\mathbb{R}^d} \|H_V(z)\|^2 + 2B^2 d^2 \leq 2B^2 dC_V^2 + 2B^2 d^2.
\end{aligned}
$$

Finally,

$$
\sum_{ijkl} |S_4(i,j)|^2 \leq \frac{2B^2}{N^2} \sum_{ijkl} \left( (\partial_k V(x_j))^2 + 1 \right) \leq 2A^2 B^2 d \left( \frac{1}{N}\sum_i V(x_i) \right)^{2\alpha} + 2B^2 d^2.
$$

Combining the above bounds, we obtain

$$
\|J\mathbf{T}(\underline{x})\|_{HS}^2 \leq 4 \left[ 2A^2 B^2 d(N+2) \left( \frac{1}{N}\sum_i V(x_i) \right)^{2\alpha} + 2B^2 d^2(N+3) + 2B^2 dC_V^2 \right].
$$

The claimed bound in the lemma now follows from the above and Lemma 3. $\qquad\square$

Now, we will complete the proof of Theorem 3.

*Proof of Theorem 3.* Fix $K > 0$ as in the theorem and sample $(x_1(0),\dots,x_N(0))$ from $p_K(0)$ supported on $\mathcal{S}_K$. Denote the law of $\underline{x}(n)$ by $p_K(n)$. As we will work with fixed $0 \leq n \leq T$ for the first portion of the proof, we will suppress dependence on $n$. Set $\nu(0) = p_K(n)$ and $\nu(\eta) = p_K(n+1)$. Interpolate these laws by defining $\nu(t) := \phi_{t\#}p_K(n)$, $t \in [0,\eta]$, where $\phi_t(\underline{x}) := \underline{x} - t\mathbf{T}(\underline{x}), t \in [0,\eta]$. Write $S_K = (\mathrm{Id} - \eta\mathbf{T})^n \mathcal{S}_K$ and $S_{K,t} := \phi_t(S_K), t \in [0,\eta]$.

Set the step-size $\eta$ as

$$
\eta = \left[ \frac{1}{C_0} \left( \frac{1}{N^{\frac{1-\alpha}{1+\alpha}} T^{\frac{2\alpha}{1+\alpha}}} \wedge \frac{1}{N} \right) \right]^{1/2} \theta, \quad \text{where} \quad \theta \in [0,1]
$$

will be appropriately chosen later and $C_0 := 2[24A^2 B^2 dM^{2\alpha}(d^{\frac{1}{1-\alpha}} + K)^{2\alpha} + 8B^2 d(C_V^2 + 4d)]$. By Lemma 4, this choice of $\eta$ ensures that for any $\underline{x} \in S_K, t \in [0,\eta]$,

$$
\|tJ\mathbf{T}(\underline{x})\|_{op} \leq \eta \|J\mathbf{T}(\underline{x})\|_{HS} \leq \frac{\theta}{2} \leq \frac{1}{2}.
$$

Thus, $J\phi_t(\underline{x})$ is invertible for any such $\underline{x}, t$ and

$$\|(J\phi_t(\underline{x}))^{-1}\|_{op} \leq \sum_{k=0}^{\infty} \eta^k \|J\mathbf{T}(\underline{x})\|_{HS}^k \leq 2.$$

In particular, if $\nu(0)$ admits a density, then for any $t \in [0, \eta]$, $\nu(t)$ admits a density given by

$$q_K(t, \underline{x}) = \left(\det(J\phi_t(\phi_t^{-1}(\underline{x})))\right)^{-1} p_K(n, \phi_t^{-1}(\underline{x})), \quad \underline{x} \in S_{K,t}$$

and $q_K(t, \underline{x}) = 0$ otherwise. Writing $E(t) := \mathsf{KL}(q_K(t)||\pi^{\otimes N})$, we obtain the following Taylor expansion on the interval $[0, \eta]$ along the lines of Korba et al. (2020):

$$E(\eta) = E(0) + \eta E'(0) + \int_0^{\eta} (\eta - t) E''(t) \mathrm{d}t. \tag{18}$$

clearly, $E(0) = \mathsf{KL}(p_K(n)||\pi^{\otimes N})$ and $E(\eta) = \mathsf{KL}(p_K(n+1)||\pi^{\otimes N})$. Moreover, by computations similar to Korba et al. (2020), writing $\hat{\nabla}V(\underline{x}) = (\nabla V(x_1), \ldots, \nabla V(x_N))'$, we obtain for $t \in [0, \eta]$,

$$E'(t) = -\int \mathrm{tr}\left((J\phi_t(\underline{x}))^{-1}\partial_t J\phi_t(\underline{x})\right) p_K(n, \underline{x})\mathrm{d}\underline{x} + \int \langle \hat{\nabla}V(\phi_t(\underline{x})), \partial_t\phi_t(\underline{x})\rangle p_K(n, \underline{x})\mathrm{d}\underline{x}$$

$$= \int \mathrm{tr}\left((J\phi_t(\underline{x}))^{-1}J\mathbf{T}(\underline{x})\right) p_K(n, \underline{x})\mathrm{d}\underline{x} - \int \langle \hat{\nabla}V(\phi_t(\underline{x})), \mathbf{T}(\underline{x})\rangle p_K(n, \underline{x})\mathrm{d}\underline{x}.$$

In particular, recalling $\Phi(z, w) := -\mathsf{k}(z, w)\nabla V(w) + \nabla_2\mathsf{k}(z, w)$,

$$E'(0) = \int \left(\mathrm{div}(\mathbf{T}(\underline{x})) - \langle \hat{\nabla}V(\underline{x}), \mathbf{T}(\underline{x})\rangle\right) p_K(n, \underline{x})\mathrm{d}\underline{x}$$

$$= \int \left(-\frac{1}{N}\sum_{i,j} \mathrm{div}_{x_i}\Phi(x_i, x_j) + \frac{1}{N}\sum_{i,j} \nabla V(x_i)\Phi(x_i, x_j)\right) p_K(n, \underline{x})\mathrm{d}\underline{x}$$

$$\leq -N\mathbb{E}_{p_K(0)}\left[\mathsf{KSD}^2(\mu_n^N||\pi)\right] + c^*d$$

along the same lines as the proof of Theorem 1. Moreover, note that for $t \in [0, \eta]$,

$$E''(t) = \psi_1(t) + \psi_2(t)$$

where, using Lemma 4 and our choice of step-size $\eta$,

$$\psi_1(t) = \mathbb{E}_{\underline{x}\sim p_K(n)}\left[\langle \mathbf{T}(\underline{x}), H_V(\phi_t(\underline{x}))\mathbf{T}(\underline{x})\rangle\right] \leq C_V \sup_{\underline{x}\in S_K} \|\mathbf{T}(\underline{x})\|^2 \leq \frac{\theta^2}{4\eta^2},$$

and

$$\psi_2(t) = \int \|J\mathbf{T}(\underline{x})(J\phi_t(\underline{x}))^{-1}\|_{HS}^2 p_K(n, \underline{x})\mathrm{d}\underline{x}$$

$$\leq \sup_{\underline{x}\in S_K} \|J\mathbf{T}(\underline{x})\|_{HS}^2 \|(J\phi_t(\underline{x}))^{-1}\|_{op}^2 \leq 4 \sup_{\underline{x}\in S_K} \|J\mathbf{T}(\underline{x})\|_{HS}^2 \leq \frac{\theta^2}{\eta^2}.$$

Combining the above observations, we obtain the following key 'descent lemma' for any $0 \leq n \leq T$:

$$\mathsf{KL}(p_K(n+1)||\pi^{\otimes N}) \leq \mathsf{KL}(p_K(n)||\pi^{\otimes N}) - N\eta\mathbb{E}_{p_K(0)}\left[\mathsf{KSD}^2(\mu_n^N||\pi)\right] + c^*d\eta + \theta^2.$$

Hence, for $T \geq 2$,

$$\mathbb{E}_{p_K(0)}\left[\frac{1}{T}\sum_{n=0}^{T-1} \mathsf{KSD}^2(\mu_n^N||\pi)\right] \leq \frac{\mathsf{KL}(p_K(0)||\pi^{\otimes N})}{NT\eta} + \frac{c^*d}{N} + \frac{\theta^2}{N\eta}.$$

Note that we have,

$$\mathsf{KL}(p_K(0)||\pi^{\otimes N}) = \int_{S_K} \frac{p_0(\underline{x})}{\mu_0(S_K)}\left[\log\left(\frac{p_0(\underline{x})}{\pi^{\otimes N}(\underline{x})}\right) - \log(\mu_0(S_K))\right]\mathrm{d}\underline{x}$$

$$\leq \frac{1}{\mu_0(S_K)}\int_{(\mathbb{R}^d)^N} p_0(\underline{x})\log\left(\frac{p_0(\underline{x})}{\pi^{\otimes N}(\underline{x})}\right)\mathrm{d}\underline{x} + \log\left(\frac{1}{\mu_0(S_K)}\right).$$

Hence, under our assumption that $K$ satisfies $\mu_0(\mathcal{S}_K) \geq 1/2$, we have that

$$\mathsf{KL}(p_K(0)||\pi^{\otimes N}) \leq 2\mathsf{KL}(p(0)||\pi^{\otimes N}) + \log 2 \leq \gamma dN,$$

where $\gamma := 2C_{KL} + \log 2$. Using this in the previous display and recalling the choice of $\eta$, we obtain

$$\mathbb{E}_{p_K(0)}\left[\frac{1}{T}\sum_{n=0}^{T-1}\mathsf{KSD}^2(\mu_n^N||\pi)\right] \leq \gamma d\sqrt{C_0}\frac{\sqrt{N^{\frac{1-\alpha}{1+\alpha}}T^{\frac{2\alpha}{1+\alpha}} \vee N}}{T\theta} + \frac{c^*d}{N} + \sqrt{C_0}\frac{\theta\sqrt{N^{\frac{1-\alpha}{1+\alpha}}T^{\frac{2\alpha}{1+\alpha}} \vee N}}{N}.$$

The above expression is 'approximately' optimized on taking $T = \lceil N^{\frac{2}{1-\alpha}}\rceil$ and $\theta = \sqrt{N/T} = N^{-\frac{1+\alpha}{2(1-\alpha)}}$, which gives the bound

$$\mathbb{E}_{p_K(0)}\left[\frac{1}{T}\sum_{n=0}^{T-1}\mathsf{KSD}^2(\mu_n^N||\pi)\right] \leq \frac{(\gamma d+1)\sqrt{C_0} + c^*d}{N}.$$

The theorem follows from the above upon noting that $\sqrt{C_0} \leq c\left(d^{\frac{1+\alpha}{2(1-\alpha)}} + \sqrt{d}K^\alpha + d\right)$ for some constant $c$ depending only on the constants appearing in Assumption 2. $\qquad\square$

## A.4 PROOF OF THEOREM 4

The additional bilinear term in equation 11 is the key to tackling Wasserstein convergence. It gives uniform control in $N, T$ over the second moment of the particle locations in the SVGD dynamics equation 2, given in the following lemma.

**Lemma 5.** *Under the same setting of Theorem 4, we have that*

$$\limsup_{N\to\infty}\ \sup_{T\geq 1}\ \mathbb{E}\left[\frac{1}{T}\int_0^T\frac{1}{N}\sum_{i=1}^N\|x_i(t)\|^2\mathrm{d}t\right] < \infty.$$

This result is proved using Lyapunov function techniques and plays a key role in the proof of Theorem 4.

*Proof of Lemma 5.* For this proof, we will abbreviate $x_i(t)$ as $x_i$. Note that, using the SVGD equations equation 2, we have

$$\begin{aligned}
\frac{\mathrm{d}}{\mathrm{d}t}\left[\frac{1}{N}\sum_{i=1}^N V(x_i)\right] = &-\left\|\frac{1}{N}\sum_{i=1}^N \nabla V(x_i)\right\|^2 - \frac{1}{N^2}\sum_{i,j}\langle x_i, x_j\rangle\langle\nabla V(x_i), \nabla V(x_j)\rangle \\
&+ \frac{1}{N}\sum_i\langle x_i, \nabla V(x_i)\rangle - \frac{1}{N^2}\sum_{i,j}\langle\nabla V(x_i), \nabla\Psi(x_i - x_j)\rangle \\
&\underbrace{- \frac{1}{N^2}\sum_{i,j}\langle\nabla V(x_i), \Psi(x_i - x_j)\nabla V(x_i)\rangle}_{\geq 0}.
\end{aligned} \tag{19}$$

The non-negativity claim above is a consequence of positive-definiteness of the kernel obtained by $(u, v) \mapsto \Psi(u - v)$. Note that

$$\begin{aligned}
\frac{1}{N^2}\sum_{i,j}\langle x_i, x_j\rangle\langle\nabla V(x_i), \nabla V(x_j)\rangle &= \sum_{\ell,\ell'=1}^d\left(\frac{1}{N}\sum_{i=1}^N x_{i,\ell}(\nabla V(x_i))_{\ell'}\right)^2 \\
&\geq \sum_{\ell=1}^d\left(\frac{1}{N}\sum_{i=1}^N x_{i,\ell}(\nabla V(x_i))_\ell\right)^2 \\
&\geq \frac{1}{d}\left(\frac{1}{N}\sum_{i=1}^N\sum_{\ell=1}^d x_{i,\ell}(\nabla V(x_i))_\ell\right)^2 \\
&= \frac{1}{d}\left(\frac{1}{N}\sum_{i=1}^N\langle x_i, \nabla V(x_i)\rangle\right)^2,
\end{aligned}$$

where the penultimate step follows by Cauchy-Schwartz inequality. Using the above inequality in equation 19, we obtain

$$\frac{\mathsf{d}}{\mathsf{d}t}\left[\frac{1}{N}\sum_{i=1}^{N}V(x_i)\right] \leq -\frac{1}{d}\left(\frac{1}{N}\sum_{i=1}^{N}\langle x_i,\nabla V(x_i)\rangle\right)^2 - \left\|\frac{1}{N}\sum_{i=1}^{N}\nabla V(x_i)\right\|^2$$
$$+\frac{1}{N}\sum_{i}\langle x_i,\nabla V(x_i)\rangle - \frac{1}{N^2}\sum_{i,j}\langle\nabla V(x_i),\nabla\Psi(x_i-x_j)\rangle. \tag{20}$$

By Assumption 3, there exists $A,\alpha,\beta,\gamma > 0$ such that

$$\langle x,\nabla V(x)\rangle \geq \alpha\|x\|^2 \quad \text{for} \quad \|x\| \geq A,$$
$$\|\nabla V(x)\| \leq \beta\|x\| \quad \text{for} \quad \|x\| \geq A,$$
$$\|\nabla\Psi\|_\infty \leq \gamma.$$

Using the above in equation 20, and defining

$$\Gamma(t) := \sum_i \|x_i\|^2 \mathbb{1}[\|x_i\| \geq A],$$

we obtain

$$\frac{\mathsf{d}}{\mathsf{d}t}\left[\frac{1}{N}\sum_{i=1}^{N}V(x_i)\right] \leq -\frac{\alpha^2}{d}\left(\Gamma(t)\right)^2 + \left(\frac{2C\beta}{d}+\beta\right)\Gamma(t) + \beta\gamma\left(\Gamma(t)\right)^{1/2} + C',$$

where the constants $C,C' > 0$ are independent of $N$ (but they depend on $A$). Thus, choosing picking a sufficiently large constant $B > 0$ (which is independent of $N$), we obtain for a constant $C_B > 0$ that, for all $T > 0$,

$$\frac{\alpha^2}{2d}\int_0^T \left(\Gamma(t)\right)^2 \mathbb{1}(\Gamma(t) \geq B)\mathsf{d}t \leq \frac{1}{N}\sum_i V(x_i(0)) + C_B,$$

$$\int_0^T \left(\Gamma(t)\right)^2 \mathbb{1}(\Gamma(t) < B)\mathsf{d}t \leq B^2 T.$$

Therefore, we obtain for all $T > 0$,

$$\frac{1}{T}\int_0^T \frac{1}{N}\sum_i \|x_i(t)\|^2 \mathsf{d}t \leq \frac{1}{T}\int_0^T \left(\Gamma(t)\right)^2 \mathsf{d}t + A^2,$$

with $\quad \frac{1}{T}\int_0^T \left(\Gamma(t)\right)^2 \mathsf{d}t \leq B^2 + \frac{C_B}{T} + \frac{1}{NT}\sum_i V(x_i(0)).$

Thus, for all $T \geq 1$ and $N \geq 1$ and for a constant $D > 0$ (which is independent of $N$),

$$\mathbb{E}\left[\frac{1}{T}\int_0^T \frac{1}{N}\sum_{i=1}^{N}\|x_i(t)\|^2\right] \leq D + \frac{1}{T}\mathbb{E}\left[\frac{1}{N}\sum_i V(x_i(0))\right].$$

By the variational representation of relative entropy, for $\delta \in (0,1)$,

$$\mathbb{E}\left[\frac{1}{N}\sum_i V(x_i(0))\right] \leq \frac{1}{\delta}\log\left(\int \exp^{\delta V(z)}\pi(\mathsf{d}z)\right) + \frac{1}{N\delta}\mathsf{KL}(p^N(0)\|\pi^{\otimes N}).$$

By Assumption 3, $\pi$ is sub-Gaussian. Hence, we have

$$\limsup_{N\to\infty}\mathbb{E}\left[\frac{1}{N}\sum_i V(x_i(0))\right] < \infty,$$

from which, the result follows. $\qquad\square$

*Proof of Theorem 4.* Recall $\mathsf{H}(t) = \mathsf{KL}(p^N(t)\|\pi^{\otimes N})$. From the general KSD bound obtained in Theorem 1, with $\tilde{\mathsf{k}}$, we obtain for every $T > 0$,

$$\frac{1}{T}\int_0^T \mathbb{E}[\mathsf{KSD}^2(\mu^N(t)\|\pi)]\mathrm{d}t \leq \frac{\mathsf{H}(0)}{NT} + \frac{1}{N^2 T}\int_0^T \mathbb{E}\left[\sum_{k=1}^N C^*\left(x_k(t)\right)\right]\mathrm{d}t,$$

where $C^*(x_i(t))$ is as defined in equation 7 with the kernel $\tilde{\mathsf{k}}$. Now note that we can obtain constant $C > 0$ such that, for any $z \in \mathbb{R}^d$,

$$\nabla_2 \tilde{\mathsf{k}}(z,z)\nabla V(z) = \langle z, \nabla V(z)\rangle \leq \|z\|\|\nabla V(z)\| \leq C(1+\|z\|^2),$$
$$\tilde{\mathsf{k}}(z,z)\Delta V(z) \leq C(\|z\|^2 + 1 + \Psi(0)),$$
$$-\Delta_2\tilde{\mathsf{k}}(z,z) = -\Delta\Psi(0).$$

Hence, for all $z \in \mathbb{R}^d$, we have that $C^*(z) \leq C_1\|z\|^2 + C_2$, for some constants $C_1, C_2 > 0$. Therefore, we have for any $t > 0$ and $N \geq 1$, that

$$\frac{1}{T}\int_0^T \mathbb{E}[\mathsf{KSD}^2(\mu^N(t)\|\pi)]\mathrm{d}t \leq \frac{\mathsf{H}(0)}{NT} + \frac{C_1}{NT}\int_0^T \mathbb{E}\left[\frac{1}{N}\sum_{k=1}^N \mathbb{E}[\|x_k(t)\|^2]\right]\mathrm{d}t + \frac{C_2}{N}.$$

Now, using Lemma 5, there exists a constant $C_4 > 0$ such that for any $T \geq 1$ and $N \geq 1$,

$$\frac{1}{T}\int_0^T \mathbb{E}[\mathsf{KSD}^2(\mu^N(t)\|\pi)]\mathrm{d}t \leq \frac{\mathsf{H}(0)}{NT} + \frac{C_4}{N}.$$

By the convexity of KSD, we have for $N \geq 1$,

$$\mathbb{E}[\mathsf{KSD}(\mu_{av}^M\|\pi)] \leq \frac{C_0}{N^{1+\sigma/2}}.$$

Hence, by Borel–Cantelli lemma, we have that $\mathsf{KSD}(\mu_{av}^M\|\pi) \overset{a.s}{\to} 0$, as $N \to \infty$. The stated Wasserstein convergence result now follows by Kanagawa et al. (2022, Thm. 3.1) taking $m = \mathrm{Id}, q_m = 1, q = 2, L = L^{(2)}$ and $\Phi = \Psi$. $\qquad\square$

## A.5 PROOF OF THEOREM 5

By Theorem 3.5 in Kanagawa et al. (2022), we have that

$$\mathsf{W}_2(\mu_{av}^M, \pi) \leq C(d)(1 \vee \mathsf{KSD}(\mu_{av}^M\|\pi)^{(1-r(d))})\mathsf{KSD}(\mu_{av}^M\|\pi)^{r(d)},$$

where, from Kanagawa et al. (2022) we have

$$r(d) = \frac{1}{3\left(\frac{4d+1}{d}\right)}\frac{1}{1+t_1} \quad \text{where} \quad t_1 = \frac{3d+1}{2} + \frac{1}{3} + \left[\frac{d+1}{d} + \frac{5}{3}\right]\nu,$$

resulting in equation 16. Define $\mathcal{E} := \left\{\mathsf{KSD}(\mu_{av}^M\|\pi) \leq \frac{C_0}{\epsilon N^{1+\sigma/2}}\right\}$. By Theorem 4 and Markov's inequality, we have that $\mathbb{P}[\mathcal{E}^c] \leq \epsilon$. On the event $\mathcal{E}^c$, for $N \geq \left(\frac{C_0}{\epsilon}\right)^{\frac{2}{2+\sigma}}$, we have

$$\mathsf{W}_2(\mu_{av}^M, \pi) \leq C(d)\left(\frac{C_0}{\epsilon N^{1+\sigma/2}}\right)^{r(d)},$$

thereby proving the claim.

## A.6 PROOF OF PROPOSITION 1

Let $\mathcal{P}(\mathbb{R}^d)$ denote the space of probability measures on $\mathbb{R}^d$, and denote by $\mathcal{P}(\mathcal{P}(\mathbb{R}^d))$ the space of probability measures on $\mathcal{P}(\mathbb{R}^d)$. Let $\mathcal{L}(\mu_{av}^M)$ denote the law of the random measure $\mu_{av}^M$ and $\delta_\pi$ denote the Dirac measure at $\pi$ in $\mathcal{P}(\mathcal{P}(\mathbb{R}^d))$.

By Lemma 5 and exchangeability,

$$\sup_{N\geq 1}\mathbb{E}\left[\int_{\mathbb{R}^d}\|x\|^2\mu_{av}^M(\mathrm{d}x)\right] = \sup_{N\geq 1}\mathbb{E}\left[\int_{\mathbb{R}^d}\|x\|^2\bar{\mu}_1^M(\mathrm{d}x)\right] < \infty. \tag{21}$$

Moreover, by Assumption 3(a), $\int_{\mathbb{R}^d} \|x\|^2 \pi(\mathrm{d}x) < \infty$. Hence, by Chaintron & Diez (2022, Theorem 3.21), we conclude that $\mathsf{W}_1(\bar{\mu}_k^M, \pi^{\otimes k}) \to 0$ if and only if $\mathcal{W}_1\left(\mathcal{L}(\mu_{av}^M), \delta_\pi\right) \to 0$ as $N \to \infty$, where $\mathcal{W}_1$ is the Wasserstein distance on the space $\mathcal{P}(\mathcal{P}(\mathbb{R}^d))$ equipped with the distance function $\mathsf{W}_1$ as defined in Chaintron & Diez (2022, Definition 3.5).

Note that $\mathcal{W}_1\left(\mathcal{L}(\mu_{av}^M), \delta_\pi\right) \leq \mathbb{E}\left[\mathsf{W}_1(\mu_{av}^M, \pi)\right]$. By Theorem 4 and Jensen's inequality,

$$\mathsf{W}_1(\mu_{av}^M, \pi) \overset{a.s}{\to} 0 \qquad \text{as} \qquad N \to \infty.$$

Moreover, observe that

$$\mathsf{W}_1^2(\mu_{av}^M, \pi) \leq \mathsf{W}_2^2(\mu_{av}^M, \pi) \leq 2 \int_{\mathbb{R}^d} \|x\|^2 \mu_{av}^M(\mathrm{d}x) + 2 \int_{\mathbb{R}^d} \|x\|^2 \pi(\mathrm{d}x)$$

and hence, by equation 21 and Assumption 3(a), $\sup_{N \geq 1} \mathbb{E}\left[\mathsf{W}_1^2(\mu_{av}^M, \pi)\right] < \infty$. In particular, $\{\mathsf{W}_1(\mu_{av}^M, \pi) : N \geq 1\}$ is uniformly integrable and thus $\mathbb{E}\left[\mathsf{W}_1(\mu_{av}^M, \pi)\right] \to 0$ as $N \to \infty$. The result follows.

