# OpenReview forum: "Improved Finite-Particle Convergence Rates for Stein Variational Gradient Descent"
_ICLR.cc/2025/Conference — ICLR 2025 Oral_

### Official Review · Reviewer_J6ce · 2024-11-02

**Soundness:** 4
**Presentation:** 4
**Contribution:** 4
**Rating:** 8
**Confidence:** 4

**Summary:**

The paper studies finite-particle convergence rates for the Stein Variational Gradient Descent (SVGD) algorithm in the Kernelized Stein Discrepancy (KSD), deriving results in both continuous and discrete times. It also provides convergence rates for the continuous-time SVGD in the L2-Wasserstein metric for a specific class of kernels.

Minor typo: in line 430, either refer to Assumption 3 in the singular or the conditions in the plural.

**Strengths:**

The paper is extremely well written, and while I have not checked all the proofs in detail, the results appear to be formally correct. The authors also do a good job explaining the literature.

The authors bypass the challenges faced by previous attempts at deriving convergence rates for SVGD in the KSD metric by working with the joint density of particle locations and the N-fold product target measure, exploiting a simple and elegant relationship related to the evolution of their KL divergence.

**Weaknesses:**

A key element in the development is the N-fold product target measure $\pi^{\bigotimes N}$. Although it's definition can arguably be derived from its name, I think it would be nice to formally define it too; pressumably, $\pi^{\bigotimes N}(x_1,\cdots,x_N) := \pi(x_1)\times\cdots\times\pi(x_N)$.

**Questions:**

Theorem 5 is derived for the Matérn-family of kernels (defined by eq. (12)) since it needs Theorem 3.5 in Kanagawa et al. (2022). However, the authors claim in the main text that the restriction to such kernels is only done for "computational clarity." Could the authors add a remark further clarifying if this means that Theorem 5 extends to the broader class of kernels defined in eq. (11)?

---

> ### Author Response · Authors · 2024-11-15
> **Thank you for your thoughtful review and positive feedback on our paper.**
>
> Thank you for your thoughtful review and positive feedback on our paper.
>
> **N-fold product target measure**
>
> Thanks for this suggestion, we have now added this definition in the notation section.
>
> **Matérn-family of kernels and Computational Clarity**
>
> In principle, one can use Theorem 3.2 from Kanagawa et al., 2024, to obtain a Wasserstein-$q$ bound based on KSD bound. However, as can be seen from the presentation of this theorem, such a result will be extremely abstract. This is a main reason why we work with the Matern kernel for obtaining tangible rates. We have reworded the discussion in the draft now to reflect our choice more accurately (see page 9).

---

### Official Review · Reviewer_uGRh · 2024-11-02

**Soundness:** 3
**Presentation:** 2
**Contribution:** 3
**Rating:** 8
**Confidence:** 3

**Summary:**

This paper performs a new analysis for Stein variational gradient descent (SVGD), achieving double exponential improvement of #particle for the continuous-time and discrete-time convergence rates in Kernel Stein discrepancy (KSD) over the best known previous result. This paper also establishes the convergence in Wasserstein distance using the bilinear Matern kernel and establishes marginal convergence and long-time propagation of chaos results for the time-averaged particle laws.

**Strengths:**

* In KSD, this paper significantly improves the particle dependence over Shi & Mackey (2024) (which is known to be the previous best result) by considering the evolution of the joint distribution of N particles.
* In Wassertein-2 distance, this is the first convergence rate, albeit showing curse of dimension.
* The discussion on related works is extensive and informative.

**Weaknesses:**

* Some notations should be further clarified, and there is not a subsection collecting all the notations which makes certain parts not that readable.
* This paper is short of some discussion on assumptions.
* Comparison with previous analysis including assumptions and rates is not that clear.
* The discussion on the convergence in Wasserstein-2 distance is weak.


For more details, see Questions below.

**Questions:**

* For time-average distribution $\frac{1}{N}\int_0^N\mu^N(t, \mathrm{d}x)\ \mathrm{d}t$, why do you integrate it from $0$ to $N$ instead of from $0$ to $T$? Since $N$ is the number of particle, the time integration range is confusing to me.
* What is $\mathcal{L}$ in Assumption 4?
* The Growth condition in Assumption 2 is essential for Theorem 3. Can you discuss some relation between this condition with some other common conditions in sampling such as log Sobolev inequality, Poincare inequality and Talagrand inequality (which is also used in some previous works on SVGD)? What kind of distributions satisfy Assumption 2?
* Also, it would be helpful to collect all related results and their corresponding assumptions in a table for a clear comparison.
* The smoothness constant in condition (c) could be interesting to the convergence rate. Why don't you include that constant in Thm 3?
* What is the relation between Wasserstein distance and KSD? Since Thm 1 doesn't exhibit any curse of dimension (CoD), it would be interesting to provide some intuition about where the CoD comes from for Wasserstein distance. Is that due to the relation between Wasserstein distance and KSD? I think the convergence in Wasserstein distance is more interesting than KSD to sampling community, and thus there should be more discussion on this result. Does Matern kernel in SVGD show better performance than some bounded kernels (Gaussian RBF) in practice (maybe do some simulation)?
* If we choose $\alpha=1/2$, as remark 3 claims, the distribution will approach Gaussian. According to Thm 3, in order to achieve the averaged $\mathrm{KSD}^2\leq\varepsilon^2$, we need to choose $N\asymp 1/\varepsilon^2$, and thus $T=O(N^4)=O(1/\epsilon^8)$. Is this terrible mixing time comparable to other sampling algorithm for Gaussian targets?

I am happy to raise my score if the authors can answer some of the questions.

---

> ### Author Response · Authors · 2024-11-15
> **Thank you for your thoughtful questions and your positive evaluation.**
>
> Thank you for your thoughtful questions and your positive evaluation. We have added a notation section for easier reference to commonly used symbols.
>
> **Why do you integrate it from $0$ to $N$?**
>
> The first two bounds in Thm 1 apply to any $N$ and $T$. From those bounds, one can obtain rates for general $N$ and $T$. However, for the KSD to converge to zero, one needs both $N$ and $T$ to jointly go to infinity. If you assume $C^*< \infty$ and $\limsup_{N \rightarrow \infty} KL (p^N(0) ||\pi^{\otimes N})/N < \infty$, then one obtains a KSD bound of $O(\frac{1}{N} + \frac{1}{T})$. Thus, taking $T=N$ gives the optimal decay rate in this upper bound.
>
> **What is $\mathcal{L}$ in Assumption 4?**
>
> $\mathcal{L}(X)$ denotes the law of the random variable $X$. We have stated this explicitly in the `Notation' section.
>
> **Growth condition in Assumption 2**
>
> As discussed in Remark 3, the growth condition essentially interpolates between the exponential and Gaussian target distribution tail behaviors as $\alpha$ varies from $0$ to $1/2$, and *does not impose any convexity* on the potential $V$.
>
> For analyzing discrete-time MCMC sampling algorithms, $V$ is typically assumed to be gradient-Lipschitz. Our growth condition required for Theorem 3 (which is focussed on the discrete-time case) is comparable to such smoothness assumptions, whereas Theorem 1 (for the continuous time case) does not require such a growth condition.
>
>  Functional inequalities (like log Sobolev and Poincare inequalities) are curvature or convexity-type conditions that are used in particular to obtain guarantees in the stronger Renyi and KL divergences for MCMC algorithms. One of the significant contributions of this work is that for the KSD results, we do not assume any convexity or functional inequality and thus our results are applicable to a wide range of potentials and kernels.
>
> However, as also pointed out by Reviewer xwur, even for the population limit SVGD (infinite particle limit), obtaining KL guarantees is an interesting and open problem. Note that existing results require the Stein log Sobolev inequality which is not as well-understood as their classical counterparts.
>
> **Table for comparison**
>
> Since there is only one prior work (Shi and Mackey, 2024) on analyzing discrete-time finite-particle SVGD, such a table will not be too meaningful in our opinion. If you still insist on adding a table, we will be happy to oblige in the camera-ready version.
>
> **Smoothness constant in Theorem 3**
>
> Good point. Actually all the constants appearing in Assumption 2 are important in quantifying the convergence rates. However, the bounds would become quite clunky and hard to parse if we made all such dependence explicit. That is why we chose to only distill out the effect of the most important parameters $d$ and $N$ in the bounds.
>
> **Wasserstein, KSD and CoD**
>
> We have added a discussion in Remark 6 providing an intuitive understanding of this phenomenon. Convergence in KSD captures convergence of expectations for a class of test functions that is much smaller than that for Wasserstein convergence (see Gorham and Mackey, 2017 and Kanagawa et al., 2022). The latter class is large enough to be highly sensitive to the effect of growing dimension, thereby exhibiting the curse-of-dimensionality.
>
> In principle, Theorem 3.2 from Kanagawa et al., 2024, could be used to get a Wasserstein bound based on KSD bound. However, Theorem 3.2 is extremely abstract and hence we work with the Matern kernel for obtaining tangible rates. A practical comparison of the different choices of kernels for SVGD algorithms, although interesting, is beyond the scope of this current work.
>
> **Gaussian mixing time**
>
> Definitely not, especially when sampling from Gaussian targets. However, we give the first poly-time complexity order for a deterministic sampling algorithm for non-parametric targets with minimal assumptions on the $V$ and $k$. For Gaussian targets and bi-linear kernels, with Gaussian initialization, Liu et al., 2024, provides more detailed results on convergence rates exploiting the parametric form (by tracking the flow of means and covariances) in comparison to the whole empirical distributions in the non-parametric case.
>
> Improving the rates of SVGD or other deterministic particle methods or proving lower bounds for such methods provides a clearer understanding of deterministic and randomized sampling. The eventual goal is to mark the territories where deterministic or randomized sampling algorithms perform better.
>
> **We sincerely hope that we have addressed many of your questions, in which case, we would greatly appreciate if you could increase your score as you see fit.**

---

> > ### Author Response · Authors · 2024-11-20
> > **Gentle reminder**
> >
> > Dear Reviewer uGRh,
> >
> > As we are getting closer to the end of the discussion period (Nov 26th), we were wondering if you could let us know if we have sufficiently addressed your questions (we would also greatly appreciate if you could increase your score as you see fit to reflect this).
> >
> > Please let us know if you have any additional questions and we will be happy to address them as much as we can before the deadline. Thank you and look forward to hearing from you.
> >
> > Best regards,
> >
> > Authors

---

> > > ### Comment · Reviewer_uGRh · 2024-11-20
> > >
> > > I thank the authors for the clarification, which addresses my concern. I will raise my score.

---

### Official Review · Reviewer_Dq3R · 2024-11-03

**Soundness:** 4
**Presentation:** 3
**Contribution:** 4
**Rating:** 8
**Confidence:** 5

**Summary:**

This paper studies the finite-particle SVGD in both discrete and continuous settings. Their results include convergence results in averaged Kernel Stein Discrepancy and Wasserstein distance.

**Strengths:**

The SVGD algorithm was well studied in practice for several years already. However, its theoretical understanding was rather limited. As well discussed by the authors, past work only tackled particular (simpler) cases discretization mechanisms of the continuous SVGD, leaving the most interesting case open. That is the finite-particle discrete-time setting. In this setting they solve this problem and obtain polynomial convergence guarantees for different types of metrics, under certain conditions.
The paper is generally well-written, although certain mathematical details are omitted.

**Weaknesses:**

### General comments

- Some of the conditions in the main results are not easy to verify. See the Questions section.
- The paper is generally well-written, although certain mathematical details are omitted.

### Mathematical comments

- The derivations starting from *line 281* are not well-explained. Why does the first term on the right-hand side vanish on second line? How is the third line derived?
- In order to obtain (17), the term $(\sum_j V(x_j)/N)^{2\alpha}$ is upper bounded  by $\sum_j V(x_j)/N$. This does not seem to be correct, even with the assumption $2\alpha < 1$.
- *line 927*. "Routine computations give...". More details on this part would ease the reading.

### Notation
- *Theorem 2*  $P(x_1(t) ∈ dx)$ is slightly vague from a mathematical point of view. Perhaps explicitly stating what is meant by this notation, would make the theorem clearer.
-  $p^N (t,\underline{z})$ is defined in Lemma 1, but later in the paper the notation is changed to $p(t,\underline{z})$.
- *notation in the proof of theorem 1* To keep notation self-explanatory I would suggest to use the number of particles in the notation for $H(0)$ in the proof of Theorem 2. That is to use $H^N(0)$, as later in the proof of the last claim you use $\sup_L H^L(0)/L$.

### typos

* *line 271* equation equation -> equation.
* *line 291* missing a period at the end of the equation.
* *line 674* Extra square bracket in the equation.
* *line 747,749* There is a $\sum$ sign missing in the first term of the right side and in the definition of $f(n)$.
* *page 8* literature literature -> literature

**Questions:**

- *Theorems 1 and 2*. Is it easy to choose an initial distribution, such that $\lim\sup_{N \to \infty} \frac{1}{N} \mathrm{KL}\left(p^{N}(0) \,\big\|\, \pi^{\otimes N}\right) < \infty$? If yes, is there a general scheme for it?
- How easy would it be to replace the averaged KSD error with the 'last iterate' KSD error. In the paper you have this type of results for some time-averaged distributions. But is it possible to avoid averaging completely as in the case of LMC?

---

> ### Author Response · Authors · 2024-11-15
> **Thank you for your meticulous reading of the paper, insightful questions and positive evaluations.**
>
> Thank you for your meticulous reading of the paper, insightful questions and positive evaluations. We have fixed the typos in the revision.
>
> **Derivations starting from line 281**
>
> Please see the updated derivation in page 6. In particular, the first term vanishes due to the regularity of the density shown in Lemma 1 and by the fact that derivative of constant is zero.
>
> **In order to obtain (17)**
>
> Good question. Note that under our assumption, we have $f(n)^{2\alpha} \leq 1+ f(n)$ and hence the claim holds as we have set $D_3 = D_1 + D_2$.
>
> **line 927: Routine computations**
>
> Thanks for the suggestion. We have now added the skipped steps.
>
> **Notation: $P(x_1(t) \in dx)$**
>
> We have defined this notation formally in the notation section in page 4.
>
> **Superscript $N$**
>
> We have followed the convention that in the statements, we use the superscript. While in the proof we drop it for notation convenience (and as it is clear from the context). We have also added this explicitly in the notation section in page 4.
>
> **Initial distribution**
>
> The condition $\underset{N \to \infty}{\limsup}~\text{KL} (p^N(0) ||\pi^{\otimes N})/N < \infty$ holds, for example, if we set the law of $\underline{x}^N(0) = (x^N_1(0), \dots, x^N_N(0))$ to be $\mu_{\circ}^{\otimes N}$, where $\mu_{\circ}$ is any probability measure on $\mathbb{R}^d$ satisfying $\text{KL}(\mu_{\circ} || \pi)< \infty$. We have added this as part of Remark 1.
>
> **Avoiding averaging completely**
>
>  As the particle dynamics are deterministic given the initial locations, the time-averaging creates the necessary `smoothing' in the absence of additional noise to enable us to prove such a result. It is a great (and open) question whether such POC results hold at the last-iterate level.

---

> > ### Comment · Reviewer_Dq3R · 2024-11-26
> >
> > Thanks for the clarifications. I maintain my score.

---

### Official Review · Reviewer_xwur · 2024-11-03

**Soundness:** 4
**Presentation:** 3
**Contribution:** 4
**Rating:** 8
**Confidence:** 3

**Summary:**

This work provides a convergence analysis of the Stein Variational Gradient Descent (SVGD) algorithm in its full formulation, i.e., using finitely many particles and in discrete time. Such a quantitative convergence proof was long sought after, ever since the algorithm was first proposed in 2016. (The only previous known result, Shi and Mackey (2024), gave a quite poor convergence rate, and had a much more involved proof.) This paper thus improves upon a long line of work reviewed in the introduction. In particular, the behavior of SVGD in the infinite-particle limit is relatively well-understood, so the main difficulty lies in the finite-particle discretization.

This work's main novel insight is presented in Equation (9) in the proof of Theorem 1, in a continuous-time context: it consists in an identity which cleanly separates the finite-particle error from the infinite-particle behavior of SVGD. The finite-particle error term in this identity can be controlled by making a mild assumption on the kernel and the target distribution (Remark 1). The discrete-time, finite-particle convergence result, Theorem 3, builds upon this insight, as well as on ideas from prior works and on new estimates (Remark 4).

The Theorems 1 and 3 establish convergence in kernelized Stein discrepancy (KSD). They are complemented by convergence results in Wasserstein distance in section 4. A long-time propagation of chaos-type result for SVGD is established in section 5.

**Strengths:**

This paper establishes quantitative convergence guarantees for finite-particle (and discrete-time) SVGD, with a much better dependency on problem parameters than the only previous known analysis. This is thus arguably the first satisfactory convergence bound, for an algorithm that has attracted considerable attention from theoreticians. So this paper's achievement is highly significant.

The main novel insight used in this paper is remarkably clean. It is also quite satisfying, in that it allows for an intuitive proof strategy.

The presentation of the paper is clear, although a bit technical from section 4 onwards.

**Weaknesses:**

Section 5 on propagation of chaos (POC) could benefit from a little bit more motivation: it is not clear why POC would be a desirable property for SVGD.

Possible typos:
- In Assumption 1(b) and Lemma 1, p_0^N needs to be C2 (not C1) for p^N(t,.) to be C2
- in Assumption 2(d), maybe = is meant to be <=
- typos on line 430

**Questions:**

- In the proof of Theorem 3, the bounds on the magnitude and the Lipschitz-ness of the update operator (Lemmas 3 and 4) grow with $n$ the iteration count. Is there any way to relate those quantities back to the distance between $\mu_n^N$ and $\pi$ instead? Intuitively, this way, the time-discretization error terms could be bounded uniformly for long times...
- Are there any results on the convergence of SVGD in KL divergence (with infinite particles and/or in continuous time)? If yes, do your results allow to improve upon them? If not, would your proof approach allow to get such a result?
- Does SVGD also exhibit long-time propagation of chaos for the "last-iterate" (as opposed to "time-averaged" in Proposition 1) occupancy measure of particle locations?

---

> ### Author Response · Authors · 2024-11-15
> **Thank you for your thoughtful review and positive evaluation.**
>
> Thank you for your thoughtful review and positive evaluation. We have addressed all the typos in the revised draft. The equality in Assumption 2(d) is indeed an equality as there is a $\sup$ on the left hand side.
>
> **Section 5 on propagation of chaos (POC)**
>
> Thanks for this suggestion. We have added a line stating that such results allow us to draw multiple i.i.d. approximate samples from the target distribution $\pi$, in comparison to a single run of traditional MCMC schemes.
>
> **Proof of Theorem 3**
>
> This is a very good question. Indeed such uniform-in-time (number of iterations) bounds would significantly improve numerous results in the paper and, in some sense, is the key challenge in SVGD. The few models for which such uniform-in-time estimates can be established involve exploiting some convexity in the driving vector field (through functional inequalities) or a gradient form representation of this vector field. In SVGD, the kernelized projection, which facilitates the particle discretization, comes at the cost of lack of convexity or a suitable gradient form. In the absence of such structure, the typically used method is Grönwall's lemma which gives exponentially growing bounds (in number of iterations) on the magnitude and the Lipschitz-ness of the update operator. Our key observation in Lemmas 3 and 4 is that the positive definiteness of the kernel can be exploited to produce polynomially growing bounds instead of exponentially growing ones. Although such bounds grow with number of iterations, it can still be used to obtain fairly good estimates on the convergence rate as displayed in Theorem 3.
>
> **Convergence of SVGD in KL divergence**
>
> No, to the best of our knowledge for standard SVGD. To translate the KSD bounds to KL bounds, one would need a good understanding of a Stein log Sobolev inequality. Although some attempts have been made before (see, for example, Duncan et al., 2023), such results seem to be out of reach (especially for $d >1$). For a regularized version of SVGD, He et al., 2024 obtained KL convergence in the infinite particle regime. But regularized SVGD leads to a different dynamics compared to SVGD although it is still a deterministic sampling algorithm.  Whether our approach can be used to obtain KL convergence in the finite-particle setup for regularized SVGD is work in progress.
>
> **Long-time propagation of chaos for the last-iterate**
>
> As the particle dynamics are deterministic given the initial locations, the time-averaging creates the necessary `smoothing' in the absence of additional noise to enable us to prove such a result. It is a great (and open) question whether such POC results hold at the last-iterate level.

---

> > ### Comment · Reviewer_xwur · 2024-11-27
> >
> > Thank you for your answer, I understand better the value of Lemma 3 now. I would suggest adding a sentence before or after equation (17) to remark explicitly that $f(n)^{2\alpha} \leq 1 + f(n)$ (as you explained in your comment to Reviewer Dq3r), as the argument is not very easy to follow otherwise. I maintain my score.

---

> > > ### Author Response · Authors · 2024-11-27
> > > **Reply**
> > >
> > > Thanks for your suggestion. Yes, we will add it in the camera-ready version directly as we are not allowed to re-upload a new PDF after Nov 26th.

---

### Meta-Review · Area_Chair_BrT6 · 2024-12-21

**Metareview:**

This work provides a convergence analysis of the Stein Variational Gradient Descent (SVGD) algorithm in its full formulation, i.e., using finitely many particles and in discrete time. Such a quantitative convergence proof was long sought after, ever since the algorithm was first proposed in 2016. (The only previous known result, Shi and Mackey (2024), gave a quite poor convergence rate, and had a much more involved proof.) This paper thus improves upon a long line of work reviewed in the introduction. In particular, the behavior of SVGD in the infinite-particle limit is relatively well-understood, so the main difficulty lies in the finite-particle discretization. This work's main novel insight is presented in Equation (9) in the proof of Theorem 1, in a continuous-time context: it consists in an identity which cleanly separates the finite-particle error from the infinite-particle behavior of SVGD. The finite-particle error term in this identity can be controlled by making a mild assumption on the kernel and the target distribution (Remark 1). The discrete-time, finite-particle convergence result, Theorem 3, builds upon this insight, as well as on ideas from prior works and on new estimates (Remark 4). The Theorems 1 and 3 establish convergence in kernelized Stein discrepancy (KSD). They are complemented by convergence results in Wasserstein distance in Section 4. A long-time propagation of chaos-type result for SVGD is established in Section 5.

Some of the key strengths:
- This paper establishes quantitative convergence guarantees for finite-particle (and discrete-time) SVGD, with a much better dependency on problem parameters than the only previous known analysis. This is thus arguably the first satisfactory convergence bound, for an algorithm that has attracted considerable attention from theoreticians. So this paper's achievement is highly significant.
- The main novel insight used in this paper is remarkably clean. It is also quite satisfying, in that it allows for an intuitive proof strategy.
- The presentation of the paper is clear, although a bit technical from Section 4 onwards. The paper is generally well-written, although certain mathematical details are omitted (much of this was fixed during the author-reviewer discussion).
- The SVGD algorithm was well studied in practice for several years already. However, its theoretical understanding was rather limited. As well discussed by the authors, past work only tackled particular (simpler) cases discretization mechanisms of the continuous SVGD, leaving the most interesting case open. That is the finite-particle discrete-time setting. In this setting they solve this problem and obtain polynomial convergence guarantees for different types of metrics, under certain conditions.
- In KSD, this paper significantly improves the particle dependence over Shi & Mackey (2024) (which is known to be the previous best result) by considering the evolution of the joint distribution of N particles.
- In Wassertein-2 distance, this is the first convergence rate, albeit showing curse of dimension.
- The discussion on related works is extensive and informative.


Some of the key weaknesses:
- some typos
- Section 5 on propagation of chaos (POC) could benefit from a little bit more motivation: it was not at first clear why POC would be a desirable property for SVGD. An explanation was added later, during the discussion period.
- Some derivations were skipped - these steps were added in the discussion period.
- Some notation seems to be undefined or hard to trace. Pointers were provided in the discussion period.

---

All reviewers eventually agreed on the score: "8: accept, good paper". The paper is unanimously praised, and as the AC, I agree with their evaluation, and recommend the paper for acceptance.

**Additional Comments On Reviewer Discussion:**

see the metareview

---

### Decision · Program_Chairs · 2025-01-22

Accept (Oral)